# Nitrogen oxides emissions from selected cities in North America, Europe, and East Asia observed by TROPOMI before and after the COVID-19 pandemic

Chantelle R. Lonsdale[1] and Kang Sun[1,2]

[1]Department of Civil, Structural and Environmental Engineering, University at Buffalo, Buffalo, NY, USA
[2]Research and Education in Energy, Environment and Water Institute, University at Buffalo, Buffalo, NY, USA

**Correspondence:** Kang Sun (kangsun@buffalo.edu)

**Abstract.** Nitrogen oxides ($NO_x$ = NO + $NO_2$) emissions are estimated in three regions in the Northern hemisphere, generally located in North America, Europe, and East Asia, by calculating the directional derivatives of $NO_2$ column amounts observed by the TROPOMI instrument with respect to the horizontal wind vectors. We present monthly averaged emissions from 1 May 2018 to 31 January 2023 to capture variations before and after the COVID-19 pandemic. We focus on a diverse collection of 54 cities, 18 in each region. A spatial resolution of $0.04°$ resolves intracity emission variations and reveals $NO_x$ emission hot spots at city cores, industrial areas, and sea ports. For each selected city, post-COVID-19 changes in $NO_x$ emissions are estimated by comparing monthly and annually averaged values to the pre-COVID-19 year of 2019. While emission reductions are initially found during the first outbreak of COVID-19 in early 2020 in most cities, the cities' paths diverge afterwards. We group the selected cities into 4 clusters according to their normalized annual $NO_x$ emissions in 2019–2022 using an unsupervised learning algorithm. All but one selected North American cities fall into cluster 1 characterized by weak emission reduction in 2020 ($-7\%$ relative to 2019) and increase in 2022 by $+5\%$. Cluster 2 contains mostly European cities and is characterized by the largest reduction in 2020 ($-31\%$), whereas the selected East Asian cities generally fall into clusters 3 and 4 with the largest impacts in 2022 ($-25\%$ and $-37\%$). This directional derivative approach has been implemented in object-oriented, open-source Python and is available publicly for high-resolution and low-latency emission estimation for different regions, atmospheric species, and satellite instruments.

## 1  Introduction

The COVID-19 pandemic, which was caused by the SARS-CoV-2 virus emerged in 2019 and its still evolving variants as of writing in 2023, has resulted in unprecedented shifts in human activities and anthropogenic emissions to the Earth's atmosphere. One of the most effective and important indicators of the post-COVID-19 emission perturbations is the emission of nitrogen oxides ($NO_x$ = NO + $NO_2$) (Levelt et al., 2022, and references therein). The dominant $NO_x$ emission source is anthropogenic fossil fuel combustion. Because of its relatively short chemical lifetime, hot spots of $NO_x$ abundance can be readily identified near the emission sources. Due to its adverse health effects, $NO_x$ is a regulated primary pollutant with significant implications for secondary ozone and $PM_{2.5}$ formation and reactive nitrogen deposition (Seinfeld and Pandis, 2016; Zhang et al., 2012).

Accurate and timely quantification of $NO_x$ emission is thus essential for environmental regulation, air quality forecasting, and improved understanding of atmospheric chemistry processes.

Bottom-up $NO_x$ emission inventories have been extensively used in atmospheric composition, climate change, and human health studies from regional to global scales (Streets et al., 2003; Crippa et al., 2018; McDuffie et al., 2020; Zheng et al., 2021). However, the bottom-up emission estimates are subject to significant and often under-characterized uncertainties that originate from the lack of knowledge of emission factors, chemical processes, and spatiotemporal proxies as well as the inconsistencies among different geographical datasets. Moreover, bottom-up emission inventories require significant effort and time to compile, leading to often years of lag time before producing results. It is especially challenging for the bottom-up approaches to represent post-COVID-19 emission changes, as both the spread of variants and the policy responses of governments worldwide have been rapidly changing.

Alternatively, satellite observations can assess $NO_x$ emissions from a top-down perspective and in a more timely manner. Substantial efforts among the research community have been devoted to characterizing the $NO_x$ emission responses in the early phase of the pandemic (Gkatzelis et al., 2021, and references therein). Satellite-observed $NO_2$ tropospheric column amounts have been used to infer post-COVID-19 $NO_x$ emission perturbations through chemical transport models (CTMs) (Miyazaki et al., 2020; Ding et al., 2020; Riess et al., 2022; Kang et al., 2022), fitting of plume dispersion or box models (Sun et al., 2021; Lange et al., 2022; Dammers et al., 2022; Xue et al., 2022; Godłowska et al., 2023; Zhang et al., 2023), and calculation of the divergence of horizontal $NO_2$ flux (the flux divergence approach hereafter, de Foy and Schauer, 2022; Dix et al., 2022; Rey-Pommier et al., 2022; Chen et al., 2023). Each approach comes with its own strengths and limitations. The CTM-based approach usually resolves emissions spatiotemporally and incorporates meteorological and chemical processes, but requires significant computation and auxiliary datasets, which hinders its agility. Analytical plume or box models are generally applied to a single source region and do not resolve the spatial distribution of emissions. The flux divergence approach has the potential to rapidly map emissions over extensive areas, whereas only annual averaged emissions have been reported in specific regions.

Inspired by the flux divergence approach, Sun (2022) proposed a unified framework capable of rapidly imaging $NO_x$ emissions using only TROPOMI level 2 products and the ERA5 global reanalysis, both of which are available within a few days of lag time. Here we coin this framework as the directional derivative approach, as the flux divergence is not explicitly calculated. Instead, the emission signal originates from the directional derivative of satellite observed column amount with respect to the horizontal wind vector. The impact of topography on emission estimation, which was neglected in the flux divergence literature, is accounted for through a similar directional derivative of surface altitude. In this work, we apply the directional derivative approach to map $NO_x$ emissions at $0.04°$ grid size over extensive regions in North America, Europe, and East Asia. We focus on 18 selected cities in each region (54 cities in total) and quantify monthly $NO_x$ emissions from 1 May 2018 to 31 January 2023. We systematically compare the emissions in 2019 as the pre-COVID-19 year with those in 2020–2022 as post-COVID-19 years. The large spatiotemporal variations of $NO_x$ emissions after 2020 in comparison with 2019 highlight the complexity of post-COVID-19 emission changes and the importance of timely and persistent observation-based constraints. The normalized annual $NO_x$ emissions from all selected cities are grouped into 4 clusters using an unsupervised learning algorithm. While the initial emission reductions during the onset of the pandemic in 2020 are ubiquitous in all clusters, the

2021 and 2022 emissions diverge significantly. The directional derivative approach has been implemented in object-oriented,
open-source Python (Sun, 2023) and is available publicly for future applications in different regions, time periods, and other
satellite instruments beyond TROPOMI.

## 2 Data

### 2.1 Data for emission calculation

Following Sun (2022), we use the TROPOMI Products Algorithm Laboratory (PAL) level 2 $NO_2$ product from 1 May 2018 to
65 14 November 2021. The operational offline product is then merged, resulting in a seamless and consistent product generated
by a single retrieval processor (version 2.3.1) (van Geffen et al., 2022a). The nadir TROPOMI level 2 pixel size was $3.5 \times 7$
$km^2$ before 6 August 2019 and updated to $3.5 \times 5.5$ $km^2$ thereafter. The equator crossing of TROPOMI is at around 13:30 local
time, but due to its ground swath width of 2600 km, the measurement's local time at the swath edges may differ by $\pm$ 1 hour
from the nadir. We use only the level 2 pixels with quality assurance values above 0.75 according to the recommendation from
70 the product Algorithm Theoretical Basis Document (ATBD) (van Geffen et al., 2022b).

Besides the $NO_2$ tropospheric vertical column density, the TROPOMI product also provides surface altitude at each level
2 pixel sampled from the GMTED2010 digital elevation model and horizontal wind at 10 m above the surface sampled from
ECMWF meteorology (Eskes et al., 2022). In addition, we sample horizontal winds at 100 m and 10 m above the surface from
the ERA5 reanalysis (Hersbach et al., 2020) spatiotemporally at TROPOMI level 2 observations.

### 2.2 Data for urban area coverage

Although the $NO_x$ emissions derived from TROPOMI-observed $NO_2$ column amounts cover all the regions, it is the urban areas
that dominate the $NO_x$ emission budget and respond most to the post-COVID-19 perturbations. Cities in different countries
and continents underwent drastically different scenarios after the onset of the pandemic. The definition of each city boundary
is often ambiguous and inconsistent among geographical regions and urban datasets. To consistently identify cities globally,
we use the Global Human Built-up And Settlement Extent (HBASE) Dataset from Landsat as the indicator of urban area
coverage (Wang et al., 2017). The HBASE dataset has a native resolution of 30 m for the target year 2010, whereas we use the
aggregated version at 1 km resolution.

Cities in this study are selected from a world city list with population and city center coordinate information (Hernández,
2022). In each region of North America, Europe, and East Asia, we focus on two subregions in the north and south, based on
latitude, climate, and proximity of city clusters. Within each subregion, we select 9 cities with the consideration of population
and location. The bounds of each region, subregion, and the names of all selected cities are shown in Figures 2, 7, and 12.

For each city, we consider the urban area coverage given by the HBASE dataset within $\pm$ 50 km in the zonal and meridional
directions from the city center coordinate as the extent of the city. This $100 \times 100$ km window covers most select cities
sufficiently with frequent inclusion of surrounding satellite cities (see Figures 5, 6, 10, 11, 15, and 16 for the extents of

individual cities). Large cities that are close together may be enveloped by the same window. For example, the Washington, DC window includes most of the area covered by Baltimore (Figure 6e), and the Wuxi window includes two similar sized cities, Changzhou and Suzhou (Figure 15g). Without attempting to separate them, we treat these cities in the same window as a single metropolitan area. However, we separate the adjacent cities at the USA-Mexico border, namely San Diego and Tijuana as well as El Paso and Juarez, because of significantly different $NO_x$ emission patterns across the country border. Additionally, the windows for Los Angeles and Dallas are extended to cover the entire Los Angeles basin and the Dallas–Fort Worth–Arlington metropolitan area, and the windows for Wuxi, Tianjin, and Busan are slightly nudged to avoid cutting off significant emission sources near the defined city edge.

## 3 Methods

### 3.1 $NO_x$:$NO_2$ ratio

As TROPOMI only observes $NO_2$ column amounts, a molar ratio between $NO_x$ and $NO_2$ is needed to derive $NO_x$ column amounts and $NO_x$ emissions. Here we use a constant $NO_x$:$NO_2$ ratio of $1.32$ as suggested in many $NO_x$ emission estimation studies (Beirle et al., 2011; Goldberg et al., 2019; Beirle et al., 2019; Dix et al., 2022; Goldberg et al., 2022; Sun, 2022; Dammers et al., 2022) for all cities. More sophisticated considerations exist, which are based on the photostationary steady state assumption and model simulated ozone concentration (Beirle et al., 2021; Lange et al., 2022) or directly from model simulated NO and $NO_2$ (Lorente et al., 2019; Zhang et al., 2023). As the main focus of this work is the relative emission changes in the pre- and post-COVID-19 years, the impact of variable $NO_x$:$NO_2$ ratio will largely cancel out. Additionally, the tropospheric mean $NO_x$:$NO_2$ ratio estimated by Beirle et al. (2021) also does not show excessive variations over the three regions included in this study. Moreover, the $NO_x$:$NO_2$ ratio can be readily updated by dividing the emissions from this study by $1.32$ and then multiplying any city- and/or season-specific value.

### 3.2 $NO_x$ emission estimation

The derivation of emissions ($E$) from satellite-observed column amounts ($\Omega$) is based on the principle of mass conservation as in the following:

$$\langle E \rangle = \langle \boldsymbol{u} \cdot (\nabla \Omega) \rangle + X \langle \Omega \boldsymbol{u_0} \cdot (\nabla z_0) \rangle + \frac{\langle \Omega \rangle}{\tau}. \tag{1}$$

Here $\langle \rangle$ is the spatiotemporal averaging operator already implemented in the physical oversampling framework (Sun et al., 2018; Sun, 2023), $z_0$ is the surface altitude from level 2 files, and $\boldsymbol{u}$ and $\boldsymbol{u_0}$ are horizontal wind vectors in the planetary boundary layer (PBL) and near the surface, respectively, represented by 100-m and 10-m winds sampled from ERA5. $X$ and $\tau$ represent the inverse scale height and vertically integrated chemical lifetime and can be inferred as linear regression coefficients using monthly or further aggregated images. The full derivation of Eq. 1 can be found in Sun (2022). Despite the similarity between Eq. 1 and its counterpart in the flux divergence literature (Beirle et al., 2019, 2021; Liu et al., 2021; Dix et al., 2022; de Foy and Schauer, 2022; Rey-Pommier et al., 2022; Veefkind et al., 2023), Eq. 1 accounts for the impacts from the

horizontal divergence of wind and topography to the estimated emission, both of which are not included in the flux divergence equation and scale linearly with column amounts, $\Omega$. The first and second terms on the right-hand side of Eq. 1 are based on the directional derivatives of the column amount ($\Omega$) and surface altitude ($z_0$) with respect to the horizontal wind vectors ($\boldsymbol{u}$ and $\boldsymbol{u_0}$). Therefore, we refer to emission estimation using this equation as the directional derivative approach.

The most important difference between the flux divergence and directional derivative approach is, at flat surface and without chemical loss, whether the emissions equal the divergence of horizontal flux ($\langle \nabla \cdot (\Omega \boldsymbol{u}) \rangle$) or a directional derivative of the column amount ($\langle \boldsymbol{u} \cdot (\nabla \Omega) \rangle$). The mathematical and physical justifications of using the directional derivative instead of the flux divergence to estimate emission are provided in detail by Sun (2022), and we further list the key assumptions made by the flux divergence and directional derivative approaches in Appendix A. In brief, we assume an altitude $z_1$ that divides the
lower troposphere where emissions are mixed within and the upper troposphere where emissions are not "felt" at the satellite pixel scale, and horizontal variability is much smaller than the lower part. Together with the incompressible flow assumption (Smits, 2000), these enable us to cancel out the wind divergence term from surface to $z_1$ ($\Omega_b (\nabla \cdot \boldsymbol{u})$, where $\Omega_b$ is the subcolumn from surface to $z_1$) with the vertical flux at $z_1$. Conceptually, the upward flux of the observed species at $z_1$ would not be due to emissions, as $z_1$ is chosen not to "feel" the emission impact; the only cause of this flux is the convergence of air in the
column below that squeezes air upwards or the divergence of air below that draws air downwards. Ultimately, this leads to the only appearance of the directional derivative term in Eq. 1, instead of the flux divergence term that can be decomposed to the sum of the directional derivative term and a wind divergence term (see Eq. A1). Moreover, this study includes in general more advanced considerations of atmospheric physical and chemical processes in comparison with previous studies, which we summarize in Appendix B.

The inverse scale height ($X$) and chemical lifetime ($\tau$) remain as key unknowns and can be estimated from observational data. At locations where emission $\langle E \rangle$ is small, Eq. 1 can be rewritten as a multi-linear regression model by neglecting the emission term:

$$\langle \boldsymbol{u} \cdot (\nabla \Omega) \rangle = \beta_0 + \beta_1 \langle \Omega \boldsymbol{u_0} \cdot (\nabla z_0) \rangle + \beta_2 \langle \Omega \rangle + \varepsilon. \tag{2}$$

Here $\beta_0$ and $\varepsilon$ represent the offset and random error in the predicted variable (i.e., $\langle \boldsymbol{u} \cdot (\nabla \Omega) \rangle$) that cannot be explained by the
linear combination of predictors (i.e., $\langle \Omega \boldsymbol{u_0} \cdot (\nabla z_0) \rangle$ and $\langle \Omega \rangle$). $\beta_1$ is an estimate of the negative inverse of scale height, and $\beta_2$ is an estimate of the negative of the first order rate constant or equivalently the inverse of chemical lifetime.

For each region, terms $\langle \boldsymbol{u} \cdot (\nabla \Omega) \rangle$, $\langle \Omega \boldsymbol{u_0} \cdot (\nabla z_0) \rangle$, and $\langle \Omega \rangle$ are calculated and saved at $0.04°$ grid size and monthly resolution. Then regressions as in Eq. 2 are conducted at a subset of grid cells for each subregion. We first focus on fitting $\beta_1$ over relatively rough terrains where $NO_x$ emissions are generally much smaller than the observational error and hence negligible. This fit can be done at a relatively high time resolution (monthly) given the high signal-to-noise ratio. $\beta_2$ is
also included in this fitting, although the results are usually very noisy. This first round of fitting is limited to grid cells with $0.001 \text{ m s}^{-1} < \langle \boldsymbol{u_0} \cdot (\nabla z_0) \rangle < 0.1 \text{ m s}^{-1}$, which represent moderately rough terrains that are abundant in all regions and subregions. In the second round, the monthly fitted $\beta_1$ in the previous round is fixed, and only $\beta_2$ is fitted in flat terrains ($\langle \boldsymbol{u_0} \cdot (\nabla z_0) \rangle < 0.001 \text{ m s}^{-1}$) that are free of strong $NO_x$ emission sources ($\langle \boldsymbol{u} \cdot (\nabla \Omega) \rangle < 1 \text{ nmol m}^{-2} \text{ s}^{-1}$) and meanwhile

characterized by moderate $NO_2$ column amount ($\langle\Omega\rangle > 2.5\times10^{-5}$ mol m$^{-2}$). To address the issue of low signal-to-noise ratio, this second round fitting is conducted over climatological months. Namely, the same months for all years are aggregated before fitting. This two-round fitting procedure is generally consistent with the study over the contiguous USA by Sun (2022). The main improvements here are that the fittings are conducted in smaller subregions and that the seasonal variations of lifetime are resolved.

Figure 1 shows the fitted $NO_x$ scale heights (top) and chemical lifetimes (bottom) for each month, although the monthly lifetime is from climatology and hence the same for different years. We caution here that the resultant scale heights and lifetimes are fundamentally fitting parameters in a multi-linear regression model (Eq. 2) that minimize the impacts of topography and chemical loss on the estimated emission. Qualitatively, the seasonality of $NO_x$ scale heights is consistent with higher PBL height in the summer than winter, and the fact that the southern subregion in North America shows significantly higher scale height than other subregions is consistent with the high PBL height over southwest USA and northern Mexico (Ding et al.,

2021; Ayazpour et al., 2023). The low scale heights in East Asia may be explained by higher levels of pollution and thus more $NO_x$ distributed near the surface. The chemical lifetimes in all subregions span a broad range and are generally longer in winter than summer. The seasonal variation of lifetime in the southern subregion in East Asia is comparable to previous lifetime estimates (Mijling and Van Der A, 2012; Shah et al., 2020), whereas the lifetimes in other subregions are significantly

higher. The most plausible explanation is that the lifetime as in Eqs 1 and 2 is integrated through the vertical column, so the free tropospheric $NO_x$ contributes more in relatively clean regions. The lifetime results in the northern subregion in Europe become unreliable in winter due to low data coverage as shown by occasional negative values. We keep using these results without modification as they do not have significant impacts on the estimated emissions.

Once the monthly $NO_x$ emissions $E$ are obtained using the monthly fitted $X$ and $\tau$, $NO_x$ emissions from each selected

city are calculated by averaging $NO_x$ emission grid cells under the geographical coverage of the city (see Sect. 2.2 for the determination of city coverage). The $NO_x$ emission grid cells are weighted by the fraction of urban area coverage during the averaging.

### 3.3 Algorithms for city clustering based on annual emissions

The monthly $NO_x$ emissions from each city (9 cities per subregion and 54 cities in total) are aggregated annually for cluster

analysis in Sect. 4.4. The normalized emissions in 2019–2022 by the 4-year mean are considered the feature for each city and clustered using the k-means clustering algorithm (Likas et al., 2003). K-means algorithm partitions a set of data points in $n$-dimensional space ($n = 4$ here, corresponding to annual emissions in 2019–2022) into $k$ clusters, where each data point belongs to the cluster with the nearest mean. The mean or centroid of each cluster is representative of the general pattern of data points in the cluster. A total number of 4 clusters are used by locating the elbow point of the sum of squared errors as a

function of the number of clusters. Additionally, we reduce the feature dimension of 4 to 2 using principal component analysis, such that each city can be projected to a 2-dimensional scatter plot as shown by Figure 17.

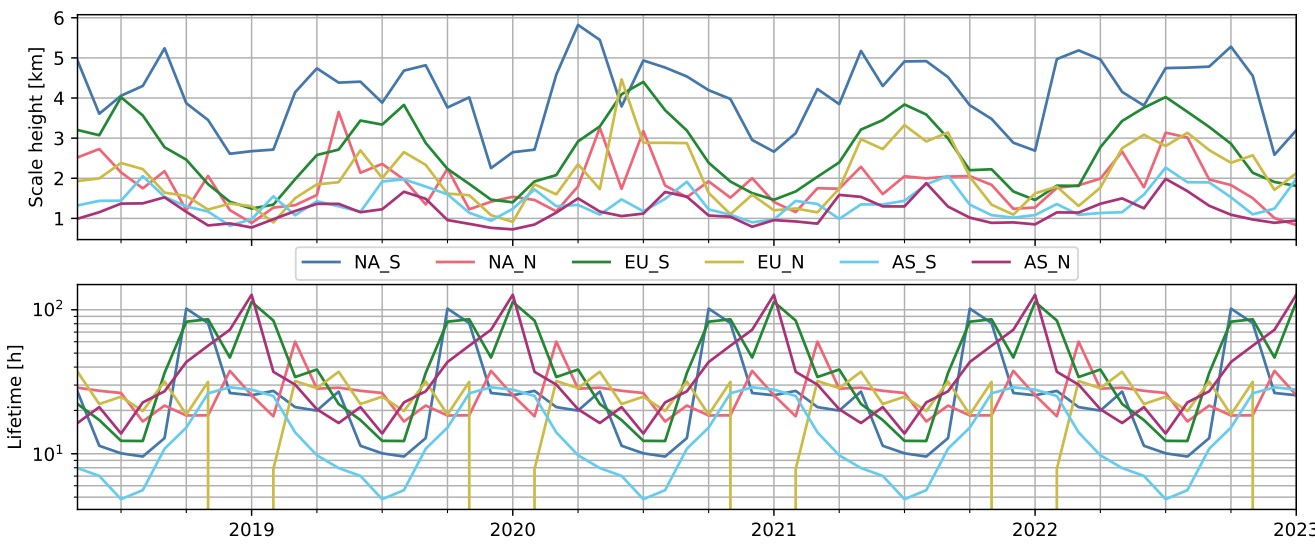

**Figure 1.** (Top) Monthly scale heights fitted using Eq. 2. NA, EU, and AS represent the regions of North America, Europe, and East Asia. S and N after the underscore denote the southern and northern subregions in each region. (Bottom) Chemical lifetimes fitted for these six subregions. Each monthly data point is from the climatology so all years have the same seasonality.

## 4  Results

This section dives into the regions in North America, Europe, and East Asia each with two subregions in the north and south and 9 selected cities in each subregion. Section 4.4 synthesizes the annual $NO_x$ emissions from all selected cities through the cluster analysis.

### 4.1  North America

Figure 2 overviews the region in North America, its two subregions delineated by red rectangles, and selected cities with locations indicated by black arrows. The spatial distribution of $NO_x$ emissions shown by the central map are estimated following Sect. 3.2 in 2019–2022, except that the scale height and chemical lifetime are fitted using the entire region instead of a specific subregion. The southern subregion covers the southwest USA and northern Mexico, and the northern subregion covers the midwest and northeast USA and part of Canada. The annual $NO_x$ emissions in 2019–2022 averaged spatially over each selected city are illustrated as pie charts around the edges of the plot. The sizes of the pies scale with the average city emissions over the 4 years. One may compare the size of slices for 2020–2022 to 2019 as an indication of post-COVID-19 emission changes. Note that the emissions in 2021 and 2022 are generally higher than 2019 for the selected cities in the southern subregion. The slices of 2019 are not larger than a quarter of the pie in all the cities in the southern subregion. Namely, despite the impacts of COVID-19, the post-COVID annual emissions in these cities are not lower than the pre-COVID year of 2019. The cities in Mexico (Tijuana, Juarez, and Chihuahua) show faster growth of $NO_x$ emissions year-to-year and stronger emissions compared

with neighboring cities in the USA. The northern subregion is quite different, in that the $NO_x$ emissions in 2019 are all higher than the 4-year average, i.e., the 2019 slices are larger than a quarter of the pies. This indicates decreased emissions after COVID-19 that may be attributed to the direct and indirect impacts of pandemic-control measures.

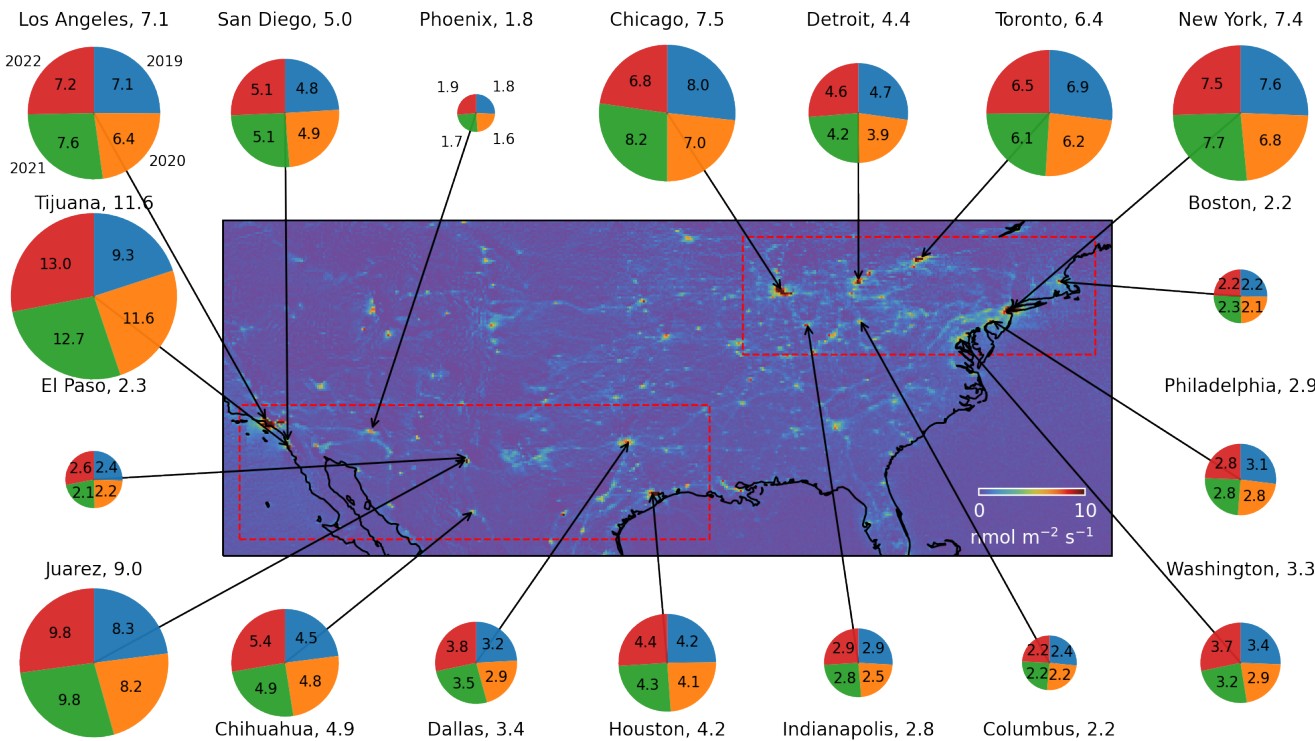

**Figure 2.** Geographical locations of the 18 cities, 9 in each of the two subregions, in the region of North America. The subregions are outlined by dashed red rectangles. The annual $NO_x$ emissions in 2019–2022 for each city are displayed as pie charts. The emission values for each year are labeled on or near the corresponding slice of the pie in nmol m$^{-2}$ s$^{-1}$. The 4-year average emissions are labeled beside the city names. The central map shows 4-year average $NO_x$ emissions throughout the region. The grid is coarsened from the native size of $0.04°$ to $0.12°$ for visualization purposes.


Figures 3 and 4 show the monthly $NO_x$ emissions that are averaged to obtain the annual emissions for cities in the southern and northern subregions in North America (see Figure 2). In these plots, each city corresponds to one panel, and the panels are ordered by population as provided by the city list (same for all the following 9-panel, 1-panel-per-city figures). For each panel, the top subpanel shows the absolute monthly $NO_x$ emissions, and the bottom subpanel shows the relative emissions to

the same months in 2019. For both subpanels, the values in 2019 are also repeated in the same months for the other years (2018 and 2020–2022) as a baseline for comparison. Higher monthly emissions relative to the same months in 2019 are indicated by red shade, and blue shade is used otherwise. We remove the monthly data with city-wide average level 2 data coverage lower than 2, i.e., the entire city area has to be on average covered at least twice in the month by TROPOMI observations. This

threshold is determined using all selected cities in this study, and monthly emissions with coverage lower than this value tend to be unreliable. To make consistent interannual comparison, if one month is removed for a particular year, the same months for all other years are also removed for a given city. This results in loss of some winter months in high latitude cities due to high solar zenith angle and snow coverage and wet-season months in some cities due to frequent cloud coverage. Although the monthly emissions are subject to significant variability, $NO_x$ emissions for most cities dropped in early 2020 relative the same months in 2019, coincident with the initial wave of COVID-19. The relative decrease is more significant in the northern subregion than in the southern subregion. The emissions, relative to 2019, diverge more in 2021 and 2022 among these cities. Some cities in the northeast USA show comparable or even more emission reduction in 2022 than in 2020 (e.g., Chicago and Philadelphia), whereas strong growth can be identified in south and southwest USA (e.g., Dallas, Houston, San Diego, and Phoenix) and in Mexico.

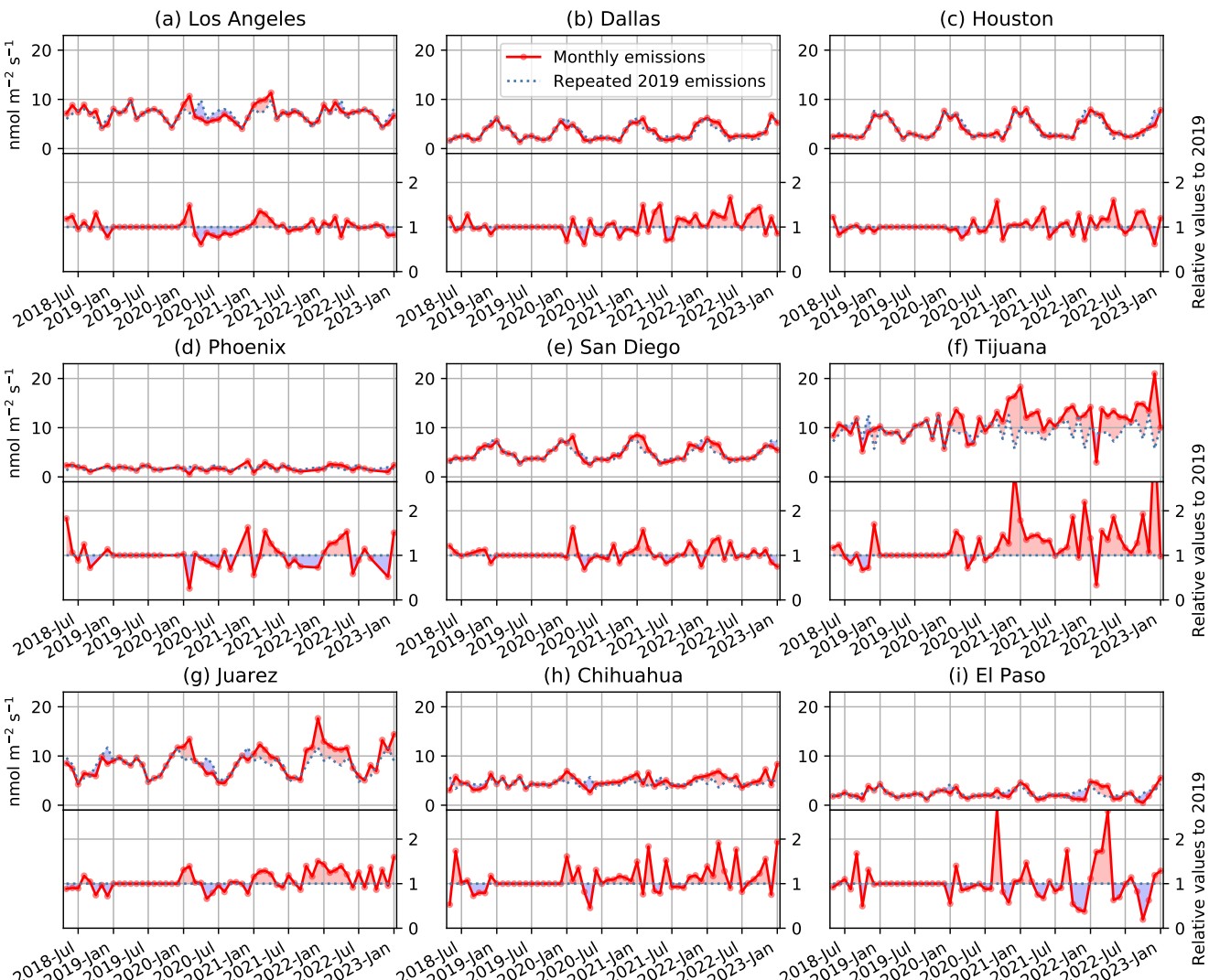

**Figure 3.** The red dots and lines show monthly $NO_x$ emissions from cities in the southern subregion in the region of North America. For each panel, the top subpanel shows the absolute emissions, and the bottom subpanel shows the relative emissions to the corresponding months in 2019. The blue dashed lines show the 2019 values repeated in the same months for the other years (2018 and 2020–2022). Red and blue shades indicate higher and lower monthly emissions relative to the months in 2019.

Strong seasonal variations with higher emissions in winter months are observed in some cites (e.g., all cities in the northern subregion, Dallas, Houston, San Diego, and Juarez), which are inconsistent with flatter seasonalities often given by bottom-up emission inventories (Sun et al., 2021). These observed seasonal variations might be caused by seasonally varying artifacts, such as retrieval biases, vertical sensitivity of the retrieval at the surface, and the uncertainties in the wind vectors. In addition, because we use a global constant $NO_x$:$NO_2$ ratio, its seasonality that is unaccounted for will propagate to the $NO_x$ emission

seasonality. One would expect higher PBL $NO_x$:$NO_2$ ratio in winter than summer, but in the summer relatively more $NO_x$ is in

the free troposphere, where $NO_x$:$NO_2$ ratio is higher than the PBL (Seinfeld and Pandis, 2016). As a result, the exact impact of

$NO_x$:$NO_2$ ratio on each city is inconclusive. However, we note that no clear seasonality can be identified in Tijuana, whereas

the adjacent San Diego shows a much more prominent seasonal pattern. This is inconsistent with the potential impacts by

the aforementioned factors, because they should have impacted the estimated city emissions similarly at such a close distance.

Moreover, similar seasonalities are not so common in the regions of Europe and East Asia to be shown in the following sections.

Further validation of the emission values and seasonality will be the subject of future studies.

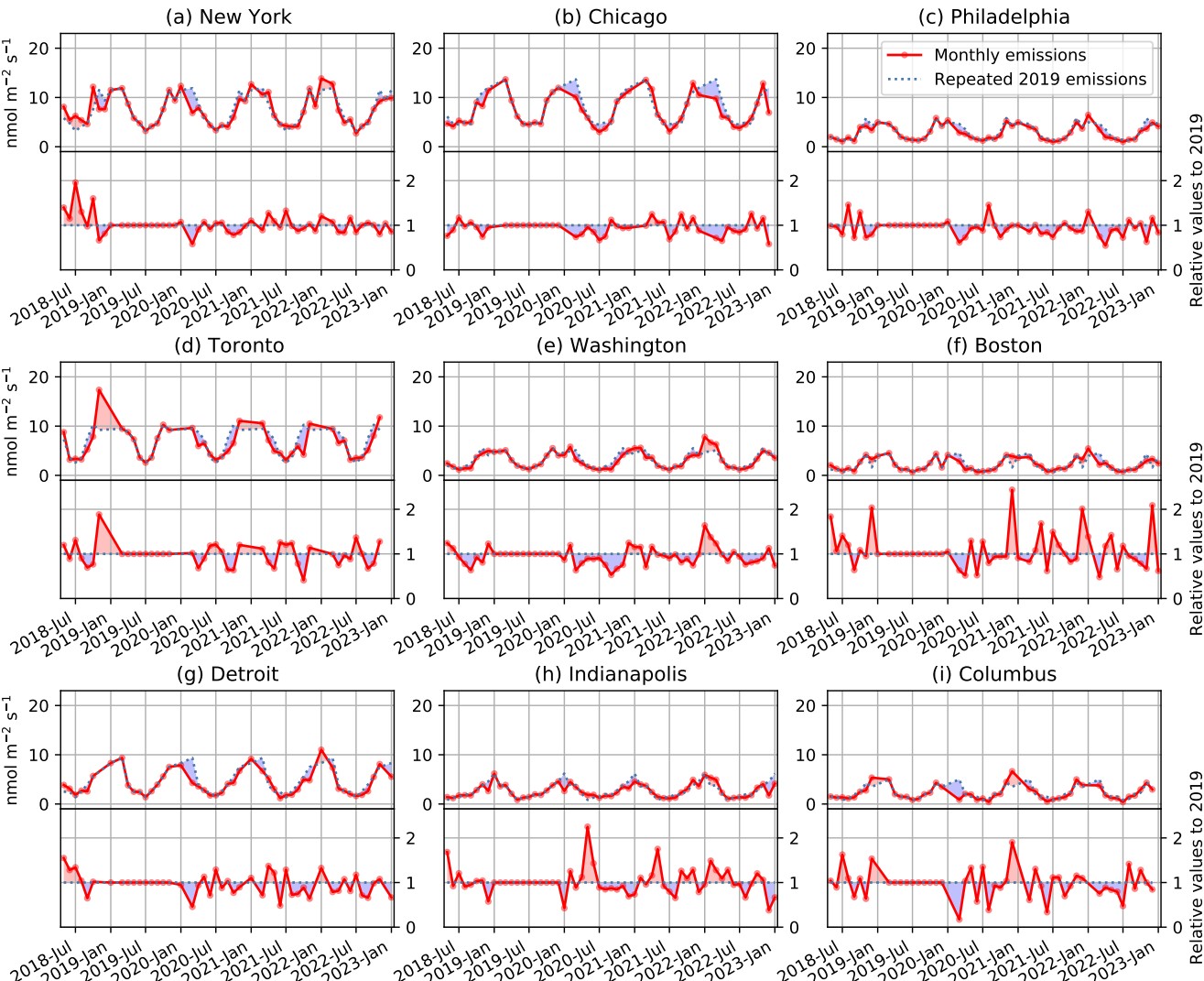

**Figure 4.** Absolute (top subpanels) and relative (bottom subpanels) monthly $NO_x$ emissions from cities in the northern subregion in the region of North America. This figure is similar to Figure 3.

The geographical urban area coverage and spatial distribution of NO$_x$ emissions for each city are shown by Figure 5 (southern subregion) and 6 (northern subregion). The averaged NO$_x$ emissions in 2019–2022 are displayed in native grid size of $0.04°$ as a colored map. The city extent is illustrated as a mask where non-urban area is in black with $95\%$ transparency, resulting in a gray hue. The city-covered area is fully transparent. The urban sprawl is significant in the USA (Barrington-Leigh and Millard-Ball, 2015), as the city areas in the USA are generally larger than similar sized cities in other countries. Emission hot spots are often collocated with the downtown areas, but industrial areas and sea ports show higher emissions, e.g., the Port of Long Beach in Los Angeles (Figure 5a) and the Houston ship channel (Figure 5c). The emissions within the city window of Washington, DC are actually dominated by Baltimore to the northeast (Figure 6e). Emissions in Tijuana and Juarez are clearly higher than the adjacent American cities San Diego and El Paso, presumably due to different emission regulations.


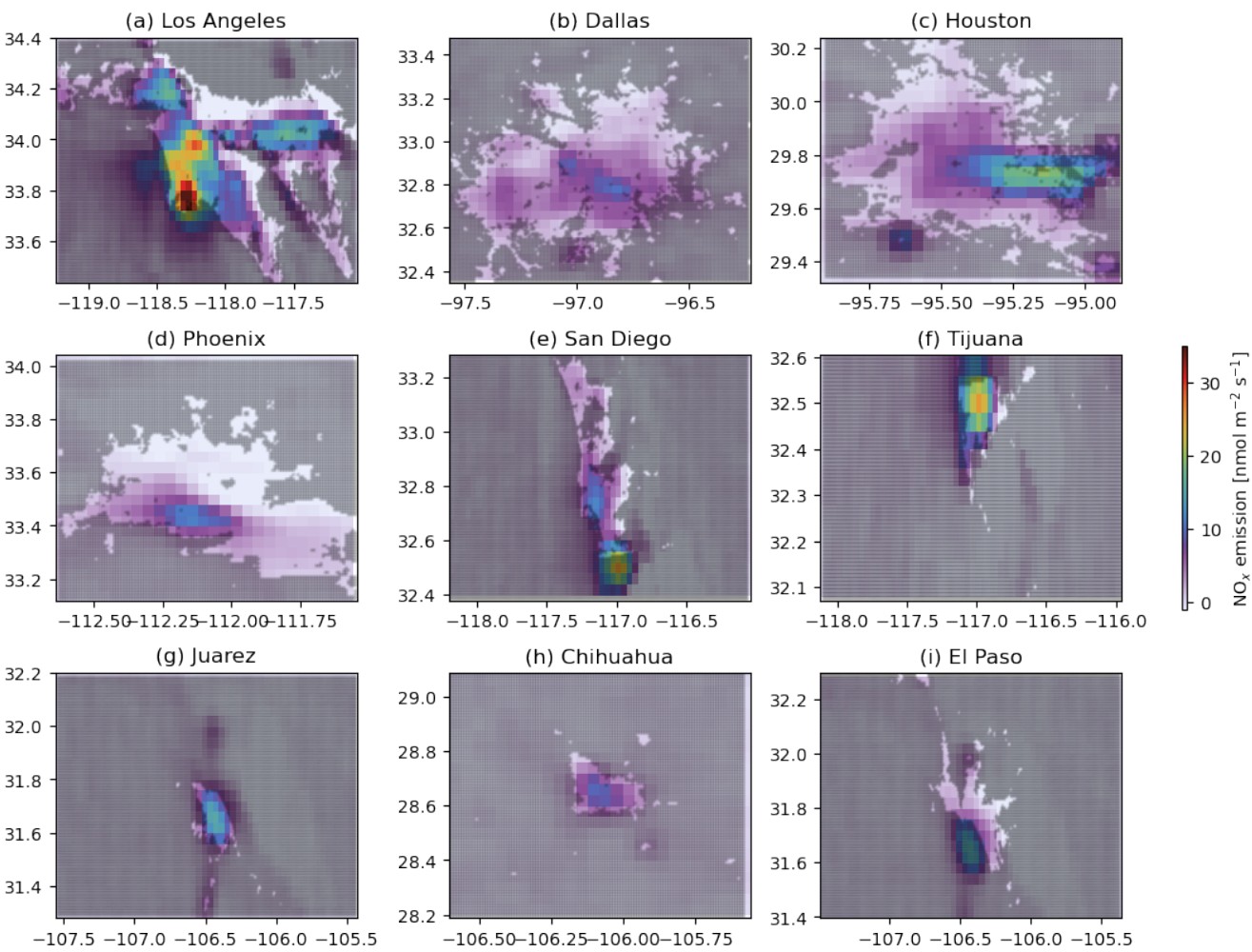

**Figure 5.** Maps of NO$_x$ emissions and urban area coverage for the 9 selected cities in the southern subregion in North America.

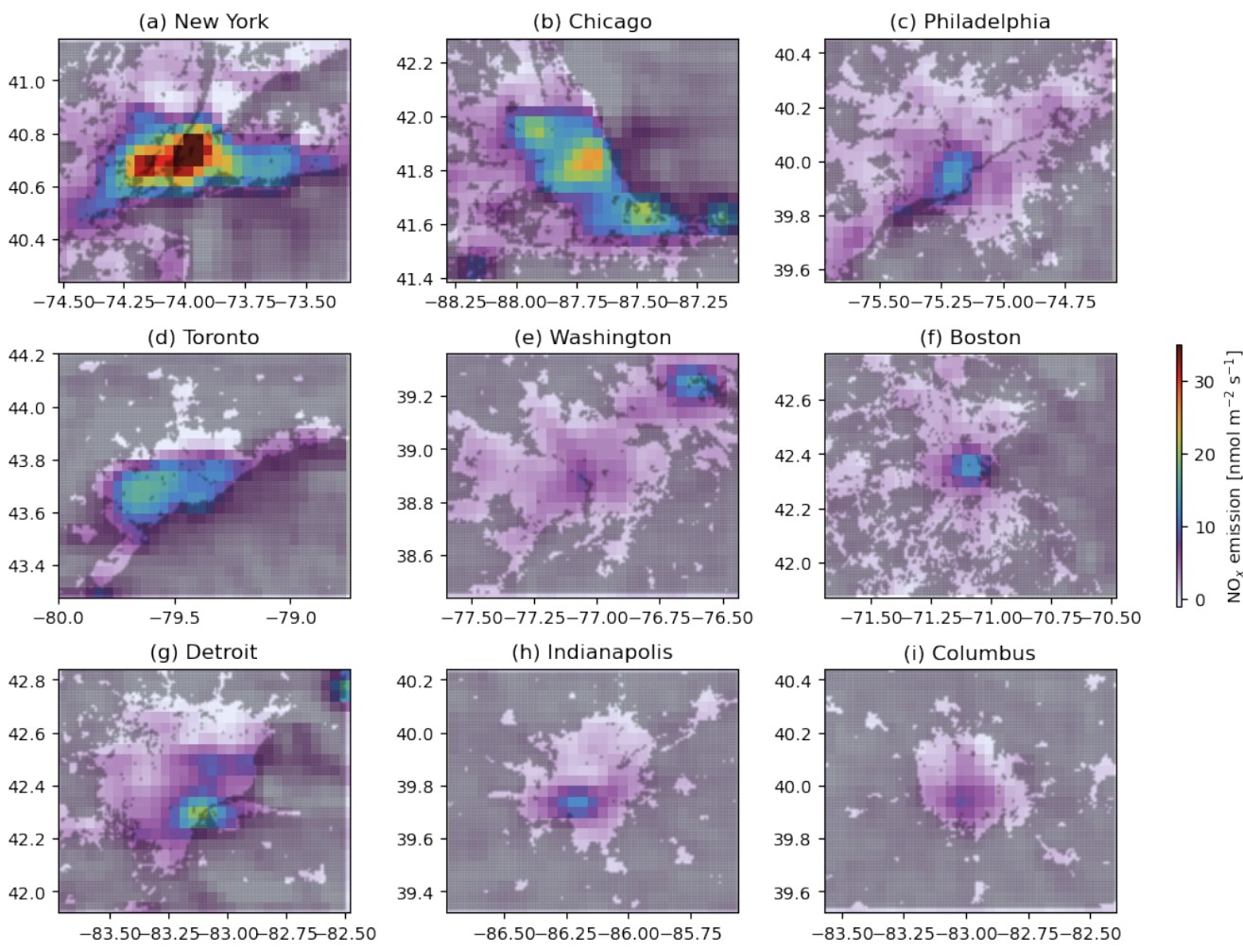

**Figure 6.** Maps of $NO_x$ emissions and urban area coverage for the 9 selected cities in the northern subregion in North America.

## 4.2 Europe

Figure 7 is a similar overview for the region of Europe, where the southern and northern subregions are delineated by red dashed rectangles. The $NO_x$ emissions from shipping lanes over the Atlantic ocean and the Mediterranean sea are prominent. The annual $NO_x$ emissions in 2019–2022 are shown similarly as pie charts for each city. Note that the southern subregion includes an African city, Algiers in Algeria, due to the rectangular shape of the subregion. Algiers also stands out among other cities in this region in that its 2019 emission is lower than the 4-year average; the 2019 slice is smaller than a quarter of the pie, and its 2020 emission is higher than 2019. All other cities show emission reductions in 2020 relative to 2019, which often extend to the following years. Two large cities in developing countries, Algiers and Istanbul, are characterized by larger emissions overall and stronger rebound of emissions after 2020.

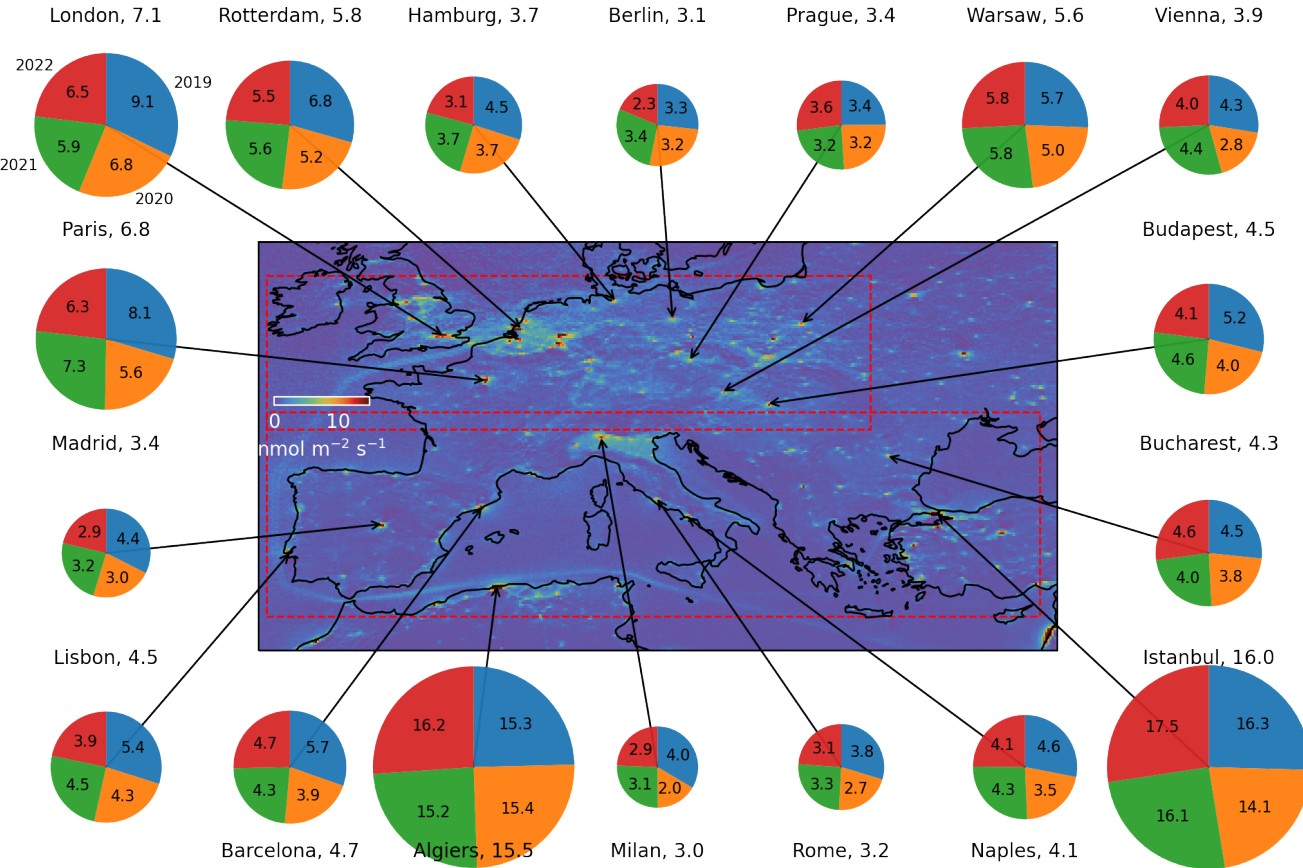

**Figure 7.** Geographical locations of the 18 cities in the region of Europe. The subregions are outlined by dashed red rectangles. The annual $NO_x$ emissions in 2019–2022 in each city are displayed as pie charts. The emission values each year are labeled on or near the corresponding slice of the pie in nmol m$^{-2}$ s$^{-1}$. The 4-year average emissions are labeled beside the city names. The central map shows 4-year average $NO_x$ emissions throughout the region. The grid is coarsened from the native size of $0.04°$ to $0.12°$ for visualization purposes.

Figures 8 and 9 show the monthly $NO_x$ emissions that are averaged to obtain the annual emissions for cities in the southern
and northern subregions in the region of Europe (see Figure 7). Compared with cities in North America, cities in the European region generally had much larger emission decreases during the initial COVID-19 wave as indicated by larger blue shaded areas. In some cases, most noticeably Madrid, Lisbon, and London, the emission reductions extend almost throughout 2020–2022. Some of the post-COVID-19 reductions relative to 2019 may extend from a pre-existing decreasing trend, as indicated by consistently higher 2018 emissions than 2019 in some cities in the northern subregion (Fig. 9). In contrast, the emissions
quickly rebounded after the initial impact in some central and eastern Europe cities, such as Bucharest, Warsaw, and Prague, as well as in Istanbul and Algiers as mentioned earlier. Some cities in the northern subregion are also subject to significant loss of winter month coverage due to high solar zenith angle and cloud coverage.

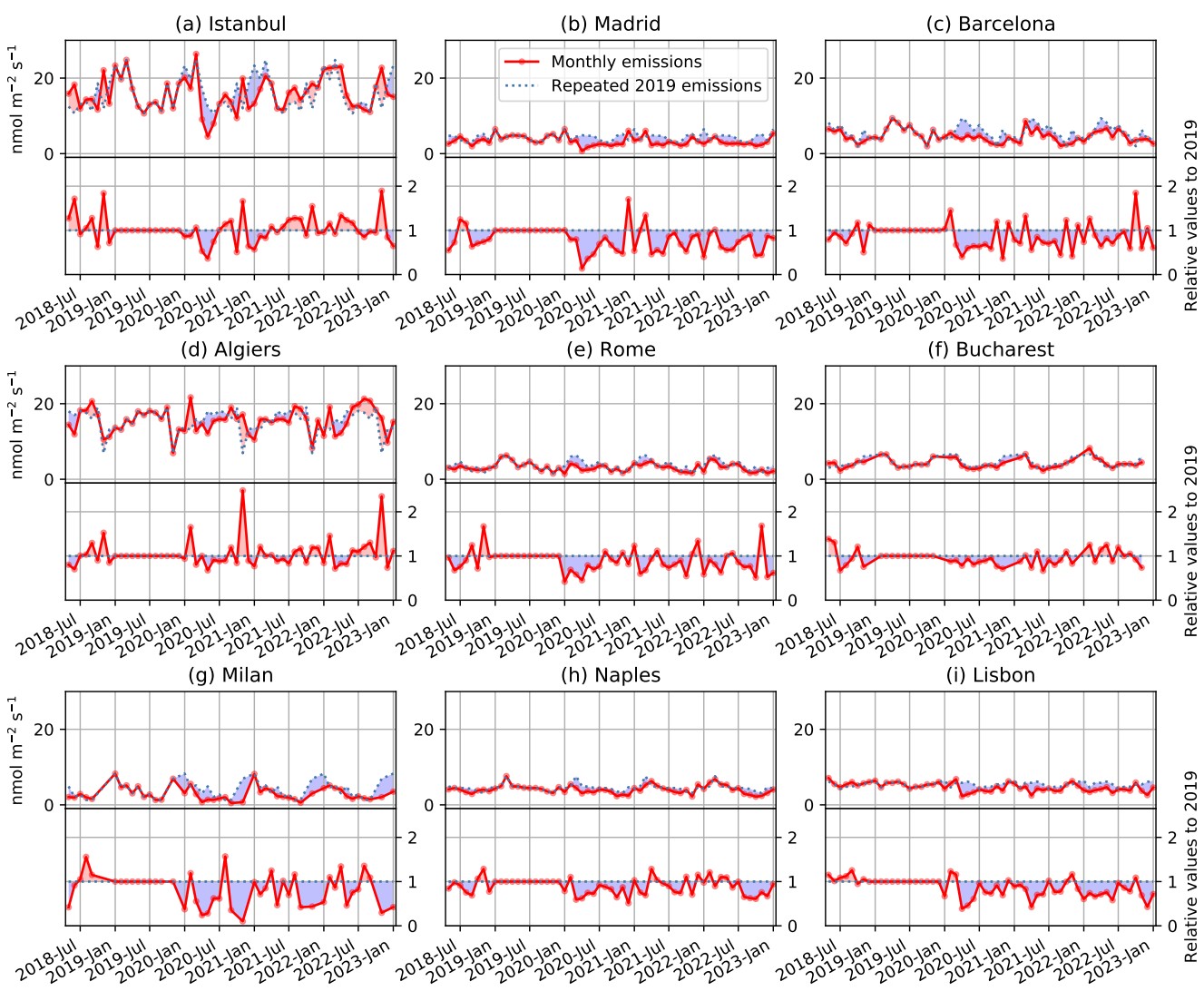

**Figure 8.** Absolute (top subpanels) and relative (bottom subpanels) monthly $NO_x$ emissions from cities in the southern subregion in the region of Europe. This figure is similar to Figure 3.

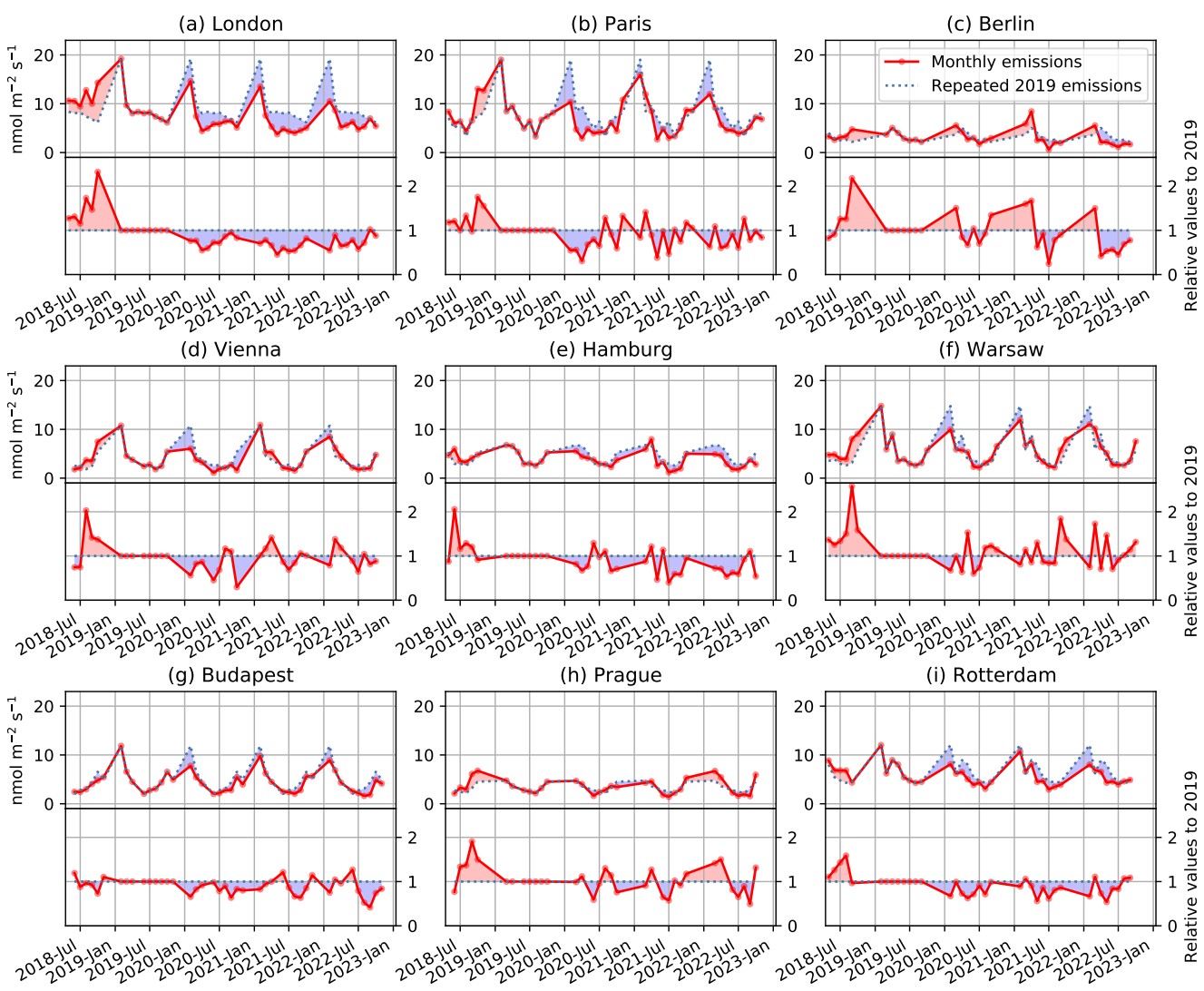

**Figure 9.** Absolute (top subpanels) and relative (bottom subpanels) monthly $NO_x$ emissions from cities in the northern subregion in the region of Europe. This figure is similar to Figure 3.

The geographical urban area coverage and spatial distribution of $NO_x$ emissions for each city are shown by Figure 10 (southern subregion) and 11 (northern subregion), similar to city maps in the North American region (Figures 5 and 6). Unlike the American cities that tend to sprawl into a large continuum, the selected European cities tend to be more concentrated with smaller satellite cities and towns scattered in the surrounding area. The most prominent emission features are generally located at the city centers, with Rotterdam as an exception where most observed emissions concentrate along the port of Rotterdam, the world's largest seaport outside of East Asia (Figure 11i).

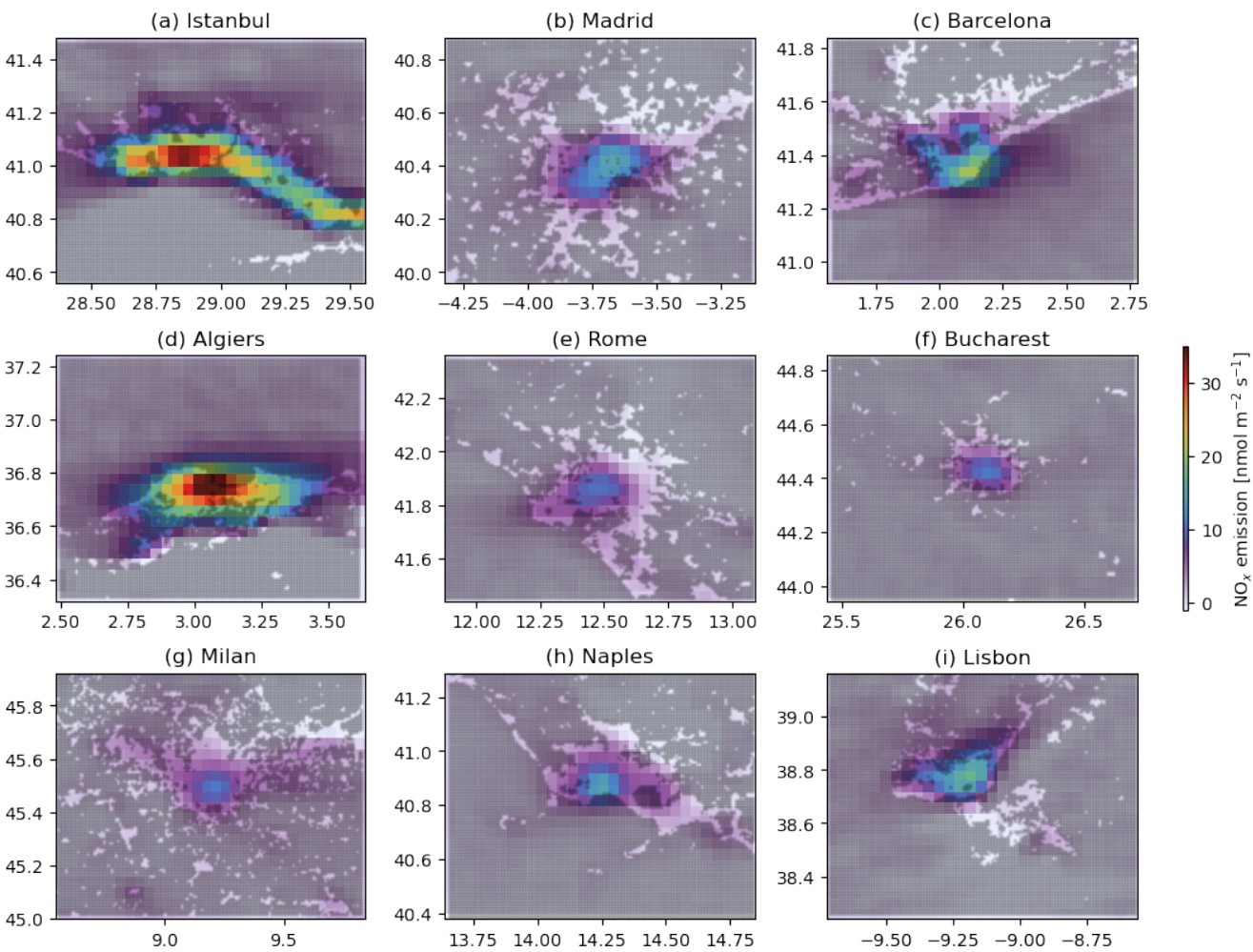

**Figure 10.** Maps of $NO_x$ emissions and urban area coverage for the 9 selected cities in the southern subregion in Europe.

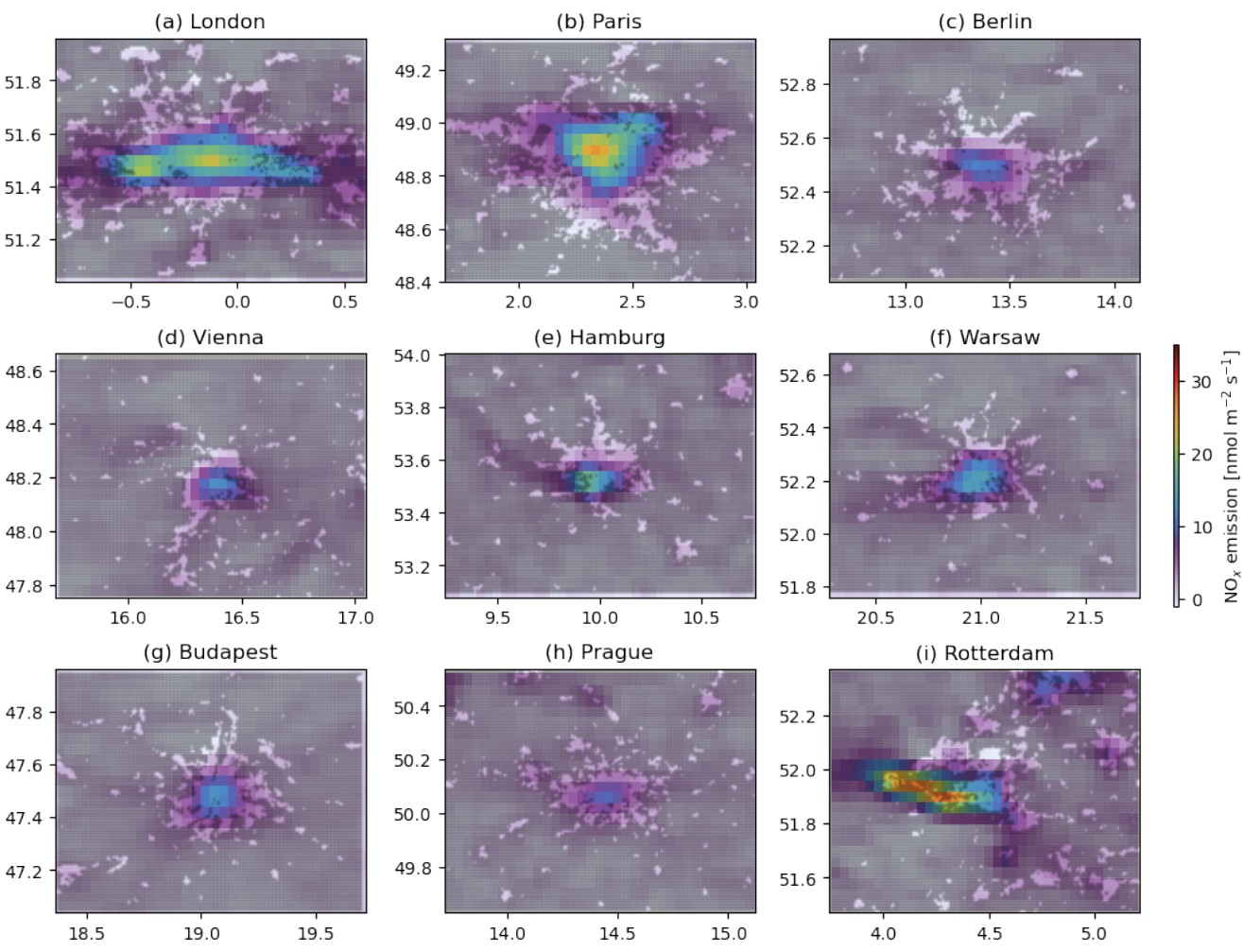

**Figure 11.** Maps of $NO_x$ emissions and urban area coverage for the 9 selected cities in the northern subregion in Europe.

## 4.3 East Asia

Figure 12 is a similar overview for the region of East Asia, where the southern and northern subregions are delineated by red dashed rectangles. Note that the overall emission background and emissions from individual cities are significantly higher than the regions of North America and Europe, as indicated by the enhanced scale of the color map and pie chart sizes. Selected cities in this region ubiquitously show lower mean annual emissions after COVID-19, i.e., the 2019 pie slices are all larger than a quarter. Unlike the regions of North America and Europe, the selected cities here all show lower emissions in 2022 than the 4-year average, and in many cases, the 2022 emissions are lower than 2021 and 2020. Out of 18 selected in this region, 16 are in China, which underwent widespread and stringent measures in 2022 to control the spread of the Omicron variant.

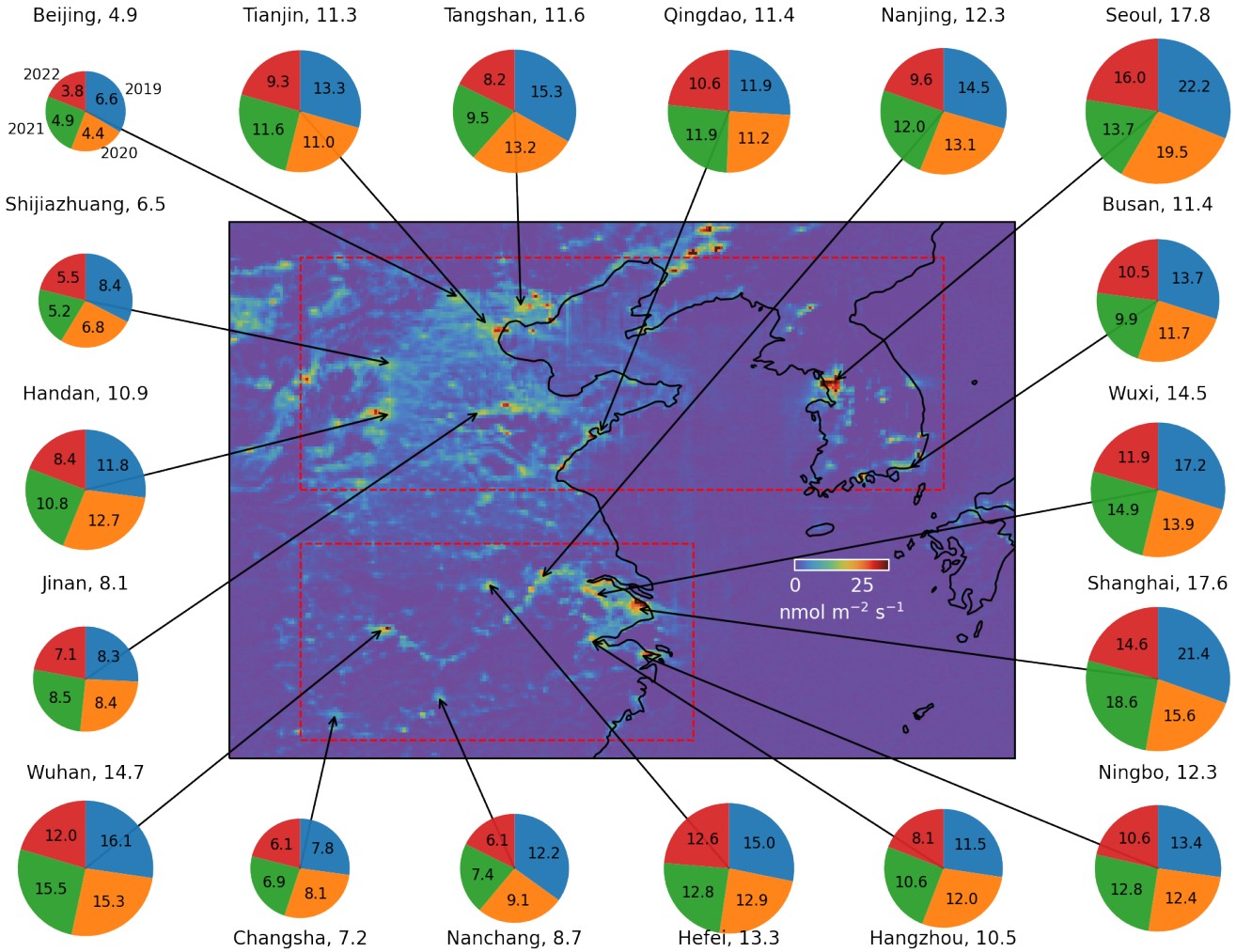

**Figure 12.** Geographical locations of the 18 cities, 9 in each of the two subregions, in the region of East Asia. The subregions are outlined by dashed red rectangles. The annual $NO_x$ emissions in 2019–2022 in each city are displayed as pie charts. The emission values each year are labeled on or near the corresponding pie slice in nmol m$^{-2}$ s$^{-1}$. The 4-year average emissions are labeled beside the city names. The central map shows 4-year average $NO_x$ emissions throughout the region. The grid is coarsened from the native size of $0.04°$ to $0.08°$ for visualization purposes.

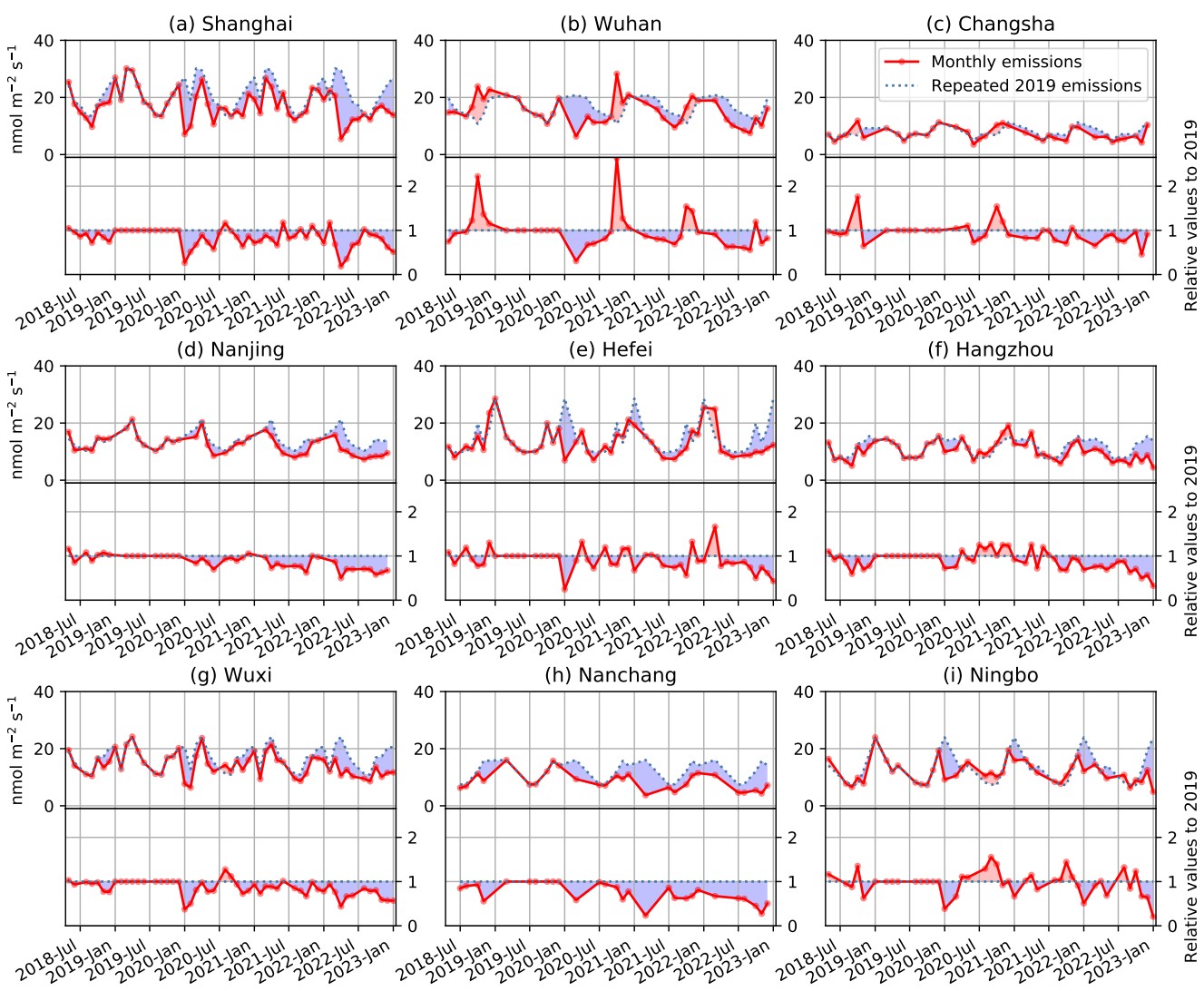

**Figure 13.** Absolute (top subpanels) and relative (bottom subpanels) monthly NO$_x$ emissions from cities in the southern subregion in the region of East Asia. This figure is similar to Figure 3.

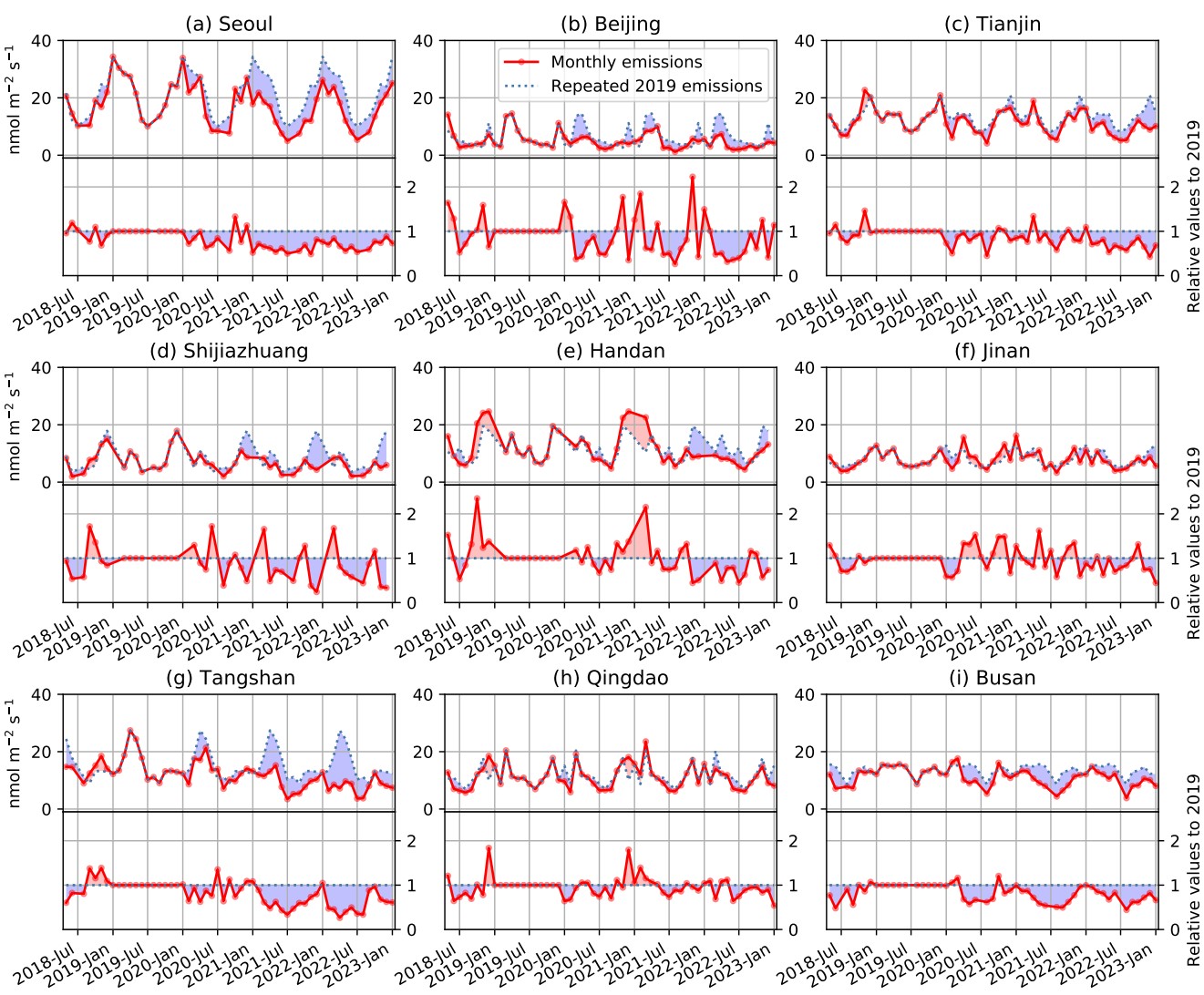

**Figure 14.** Absolute (top subpanels) and relative (bottom subpanels) monthly $NO_x$ emissions from cities in the northern subregion in the region of East Asia. This figure is similar to Figure 3.

The temporal evolution of $NO_x$ emissions and their relative changes to 2019 are shown in more detail in the monthly plotted Figures 13 and 14, with one of the strongest examples of this shown in the megacity of Shanghai in Figure 13a. Shanghai is one of the largest cities studied here with higher $NO_x$ emissions than all other selected cities except Seoul. The data coverage for Shanghai is also complete without any missing months. The well-documented $NO_x$ emission reductions during the spring festival in January and/or February are evident in 2019–2021, although the 2020 spring festival coincided with the initial control measures at the beginning of the COVID-19 outbreak (Liu et al., 2020; Huang and Sun, 2020). Emissions were back to 2019 levels during the second half of 2021. March 2022 marked the start of an unprecedented lockdown in Shanghai in response to the spread of the Omicron variant, and the resultant $NO_x$ emission reductions overshadow those of early 2020. Emission levels largely recovered in August–October 2022 before plunging again due to a nation-wide spread of the Omicron variant, which ultimately led to a termination of most control measures in China in December 2022. The full effect of this policy change on $NO_x$ emissions is not yet clear from the current data.

Figure 13b shows the monthly $NO_x$ emissions in Wuhan, where COVID-19 first drew public attention in January 2020. Data in January and February of all years are not included due to insufficient TROPOMI coverage, resulting in missing peak lockdown months. Therefore, the relative reduction in 2020 is likely underestimated. Another noteworthy feature in Wuhan is that the October emissions in 2018 and 2020–2022 are all higher than October 2019. One likely cause is the emission reduction measures conducted to ensure good air quality during the 7th Military World Games held in Wuhan in October 2019 (Zhang et al., 2022). As a result, October 2019 emission in Wuhan was likely lower than the business-as-usual condition, leading to spurious enhancements in October of all other years. The other selected cities similarly show emission reductions in early 2020 during the onset of the pandemic and more extensive reductions in 2022 due to direct and indirect impacts of the Omicron variant spread. Some cities show consecutive months of recovery or increase of emissions in between, e.g., Hangzhou, Ningbo, and Jinan, for the summer and fall of 2020.

Similar to the regions of North America and Europe, the geographical urban area coverage and spatial distribution of $NO_x$ emissions for each city in the region of East Asia are shown by Figure 15 (southern subregion) and 16 (northern subregion). Unlike most cities in North America and Europe, the strongest emissions in the selected Asian cities are often not located at the city centers, but in industrial areas or sea ports. Here we try to identify the most prominent emission hot spots on the city maps. The strongest emissions in Shanghai occur in the highly industrialized Baoshan and Pudong districts along the Yangtze River shore (Figure 15a). For Wuhan, the strongest emissions occur at Qingshan district where Wuhan Iron & Steel is located (Figure 15b). The emission hot spot in the southwest of Nanjing (Figure 15d) is the city of Maanshan, home of Maanshan Iron & Steel. The emission hot spot in the northeast of Wuxi (Figure 15g) is part of Zhanjiagang, a county-level city (in contrast to prefecture-level cities) under the administration of Suzhou. The emission hot spot to the east of Ningbo city (Figure 15i) appears to be part of the port of Ningbo-Zhoushan, the world's largest cargo handling port.

The city window around Seoul (Figure 16a) includes most of the Seoul metropolitan area, which was mapped for high-resolution $NO_2$ column amounts during the Korea–United States Air Quality (KORUS-AQ) campaign (Choo et al., 2023). The west-east extended hot spot in the west of the city window is associated with the Incheon industrial complex, while the more south-north extended hot spot at the center of the city window is located over the city of Seoul. The emission hot spot in the

south is located near Suwon, which is also industrialized. The emission hot spot to the southwest of the Tianjin city (Figure 16c) collocates with the port of Tianjin, the largest port in Northern China and the main maritime gateway to Beijing, and the adjacent industrial area in the Binhai New Area. The emission hot spot to the northeast of Tangshan city (Figure 16g) appears to be Qian'an, a county-level city under the administration of Tangshan, and is the location of the Yanshan Iron & Steel. The emission hot spot to the southwest of Qingdao city (Figure 16h) appears to be the Port of Qindao. The emission hot spot to the northeast of the Busan city (Figure 16i) appears to be the port of Ulsan, the largest industrial port in South Korea.

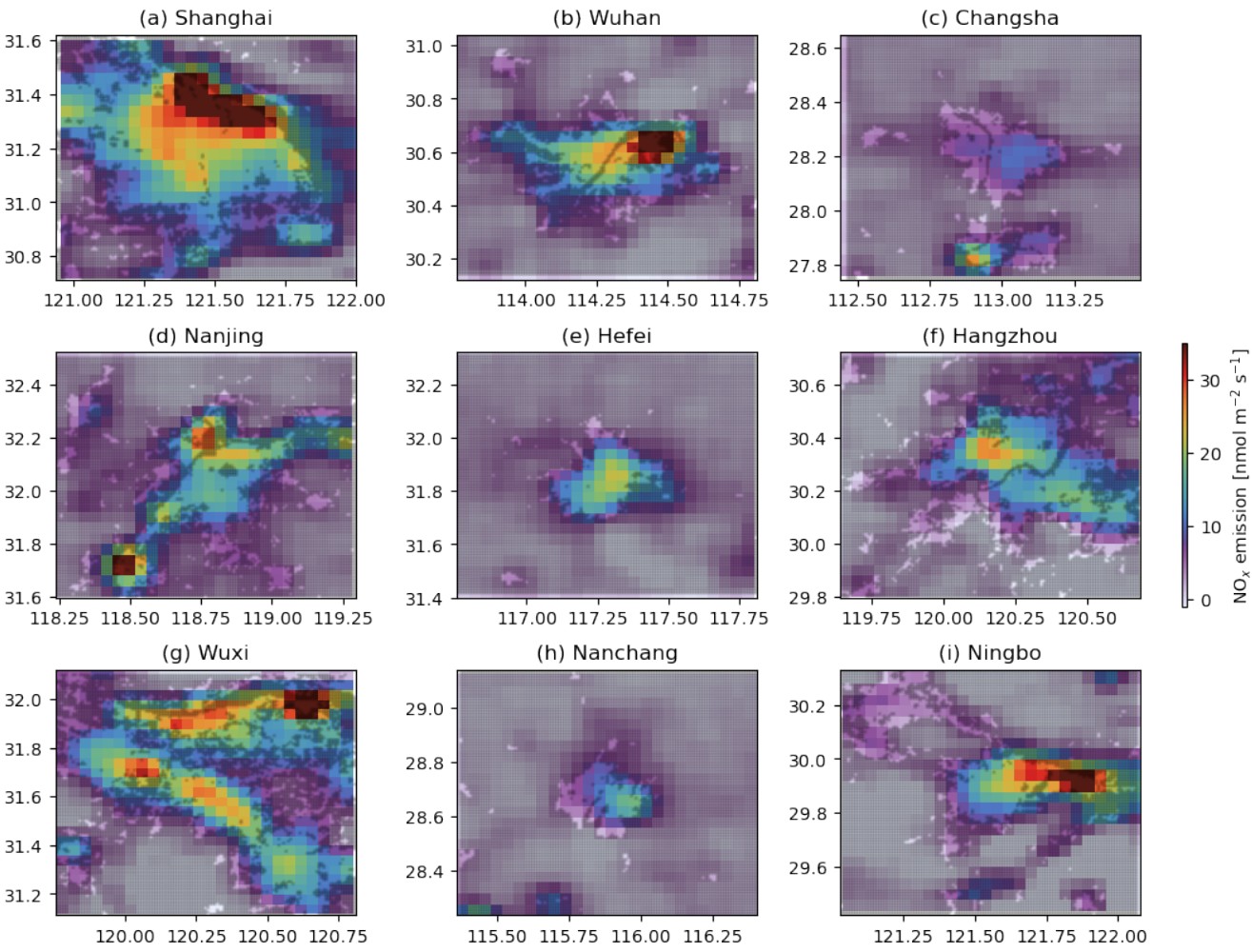

**Figure 15.** Maps of $NO_x$ emissions and urban area coverage for the 9 selected cities in the southern subregion in East Asia.

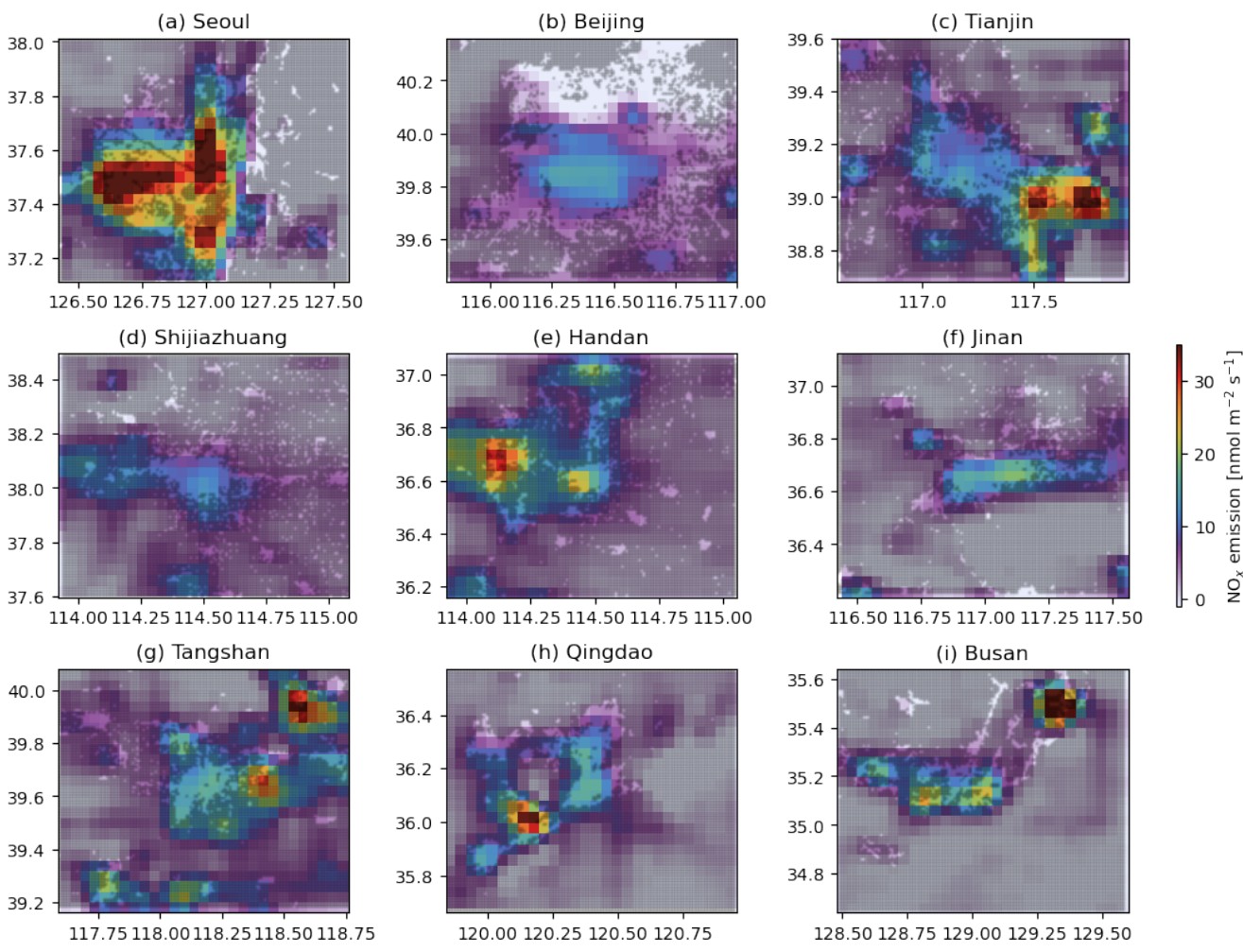

**Figure 16.** Maps of NO$_x$ emissions and urban area coverage for the 9 selected cities in the northern subregion in East Asia.

## 4.4 Clustering of city emissions

The monthly NO$_x$ emissions calculated for each of the 54 selected cities contain a large amount of information that is challenging to digest. These monthly values are often subject to low signal-to-noise ratio, especially for the cold season months in high-latitude cities. As such, we aggregate to annual emissions for the years of 2019–2022 to obtain more insights into how different cities' emissions responded after the onset of COVID-19. For each city, the same months are included for all these years to enhance interannual consistency. The resultant annual emissions are already shown in Figures 2, 7, and 12. These annual emissions (4 values for each city) are normalized by the 4-year mean and then grouped into 4 clusters using the k-means algorithm. The normalized annual emission for each city corresponds to a point in the 4-dimensional space. To visualize the

clustering results, we reduce the dimension of the normalized annual emissions by calculating 2 principal components, which effectively projects the data to a 2-dimensional subspace.

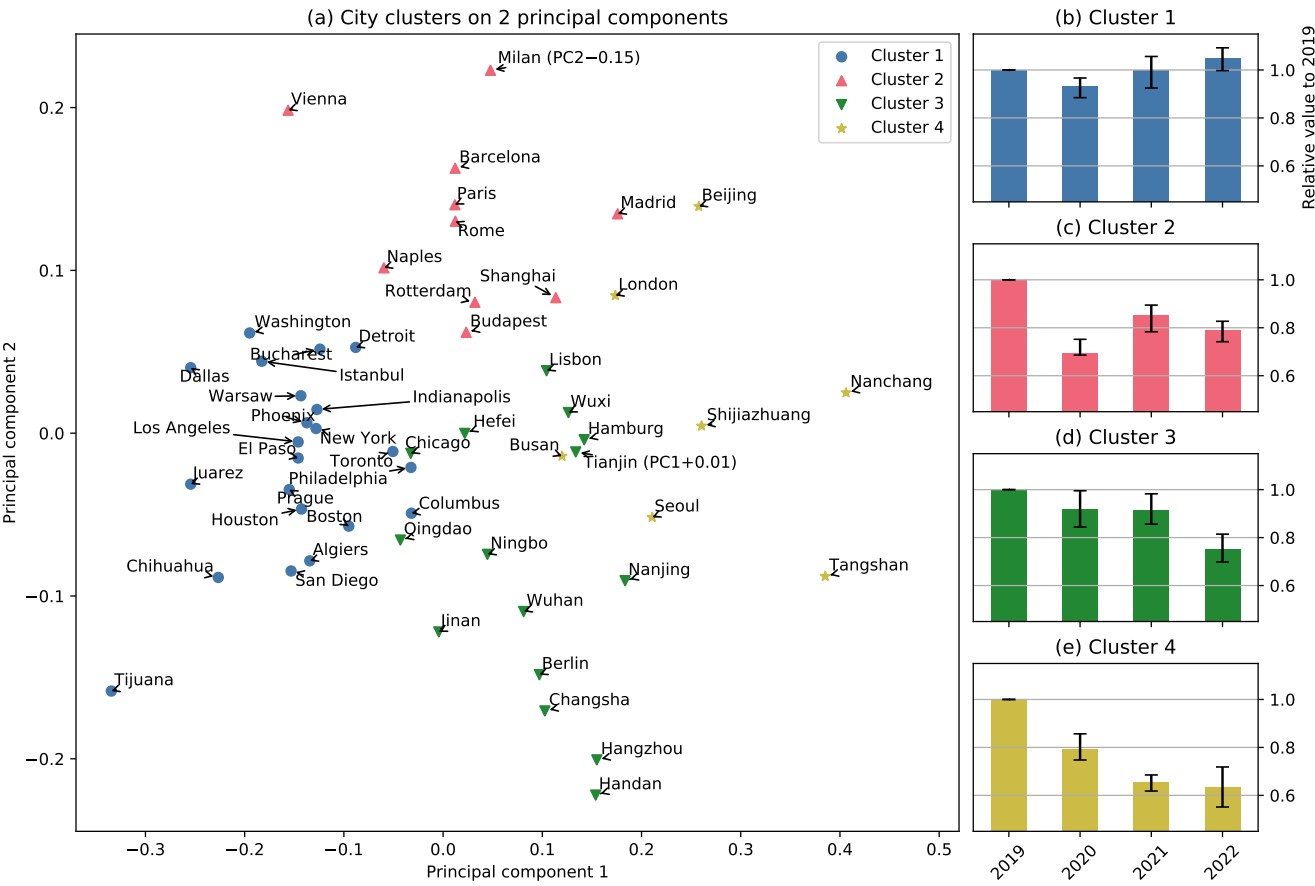

**Figure 17.** (a) Clustering results using 54 cities selected in this study. The normalized annual emissions in 2019–2022 for each city are projected to 2-dimensional space using principal component analysis. The locations of Tianjin and Milan, defined as the two principal components, are nudged downward and right, respectively, to enhance visualization quality. (b–e) The bars indicate the cluster center, and error bars indicate the interquartile range within the cluster. The values are all relative to 2019.

Figure 17a shows the distributions of cities in the projected 2-dimensional subspace, where each city is a point marked differently according to the cluster it belongs to. The annual emissions for cities in the same cluster are shown as relative values to the 2019 emission in Figure 17b–e, where the bars indicate the cluster average, and error bars indicate the interquartile range within the cluster. All cities in the North American region, with Chicago as the only exception, are included in cluster 1 (Figure 17b), which is characterized by the least emission reduction in 2020 ($-7\%$ relative to 2019), with emissions recovering to the 2019 level in 2021, and increased emissions in 2022 ($+5\%$ relative to 2019). Bucharest, Warsaw, Prague, Algiers, and Istanbul

in the European region are also included in this cluster. The overall characteristic of cluster 1 is a minor reduction in 2020 relative to 2019 and steady increase afterwards.

Cluster 2 (Figure 17c) features the most significant emission reduction in 2020 ($-31\%$ relative to 2019), with a moderate rebound in 2021 ($-15\%$), and drop again in 2022 ($-21\%$). Cities in this cluster are all in the European region, except Shanghai. Cluster 3 (Figure 17d) differs from cluster 2 in that the emission reduction in 2020 is not as large ($-8\%$), while the reduction in 2022 is very significant ($-25\%$), especially in comparison with its value in 2021 ($-8.5\%$). Cluster 3 mainly includes Cities in China with a few exceptions in Europe (Lisbon, Hamburg, and Berlin) and North America (Chicago). This is consistent with the general evolution pattern of COVID-19 in China: quick recovery of emissions in later 2020 due to effective COVID-19 control measures, sporadic lockdowns in 2021, and much larger scale lockdowns in 2022.

Cluster 4 (Figure 17e) is characterized by the largest and sustained decrease of emissions from 2019 to 2022. The average emission reduction in 2022 in this cluster is $-37\%$ relative to 2019, the lowest for all years in all clusters. The reduction in 2021 is $-35\%$, also substantially more than all other clusters. Cities in this cluster are located in northern China and South Korea with London as the only exception. Additionally, Tangshan, Seoul, and Busan in cluster 4 have large contributions from industrial $NO_x$ sources (see Figure 16), suggesting influences from economic factors besides direct COVID-19-control impacts.

## 5    Conclusions and discussion

We apply the directional derivative approach developed by Sun (2022) to estimate $NO_x$ emissions in three northern hemisphere regions; North America, Europe, and East Asia. For each region, the $NO_x$ scale heights and chemical lifetimes, which are necessary in the emission calculation, are estimated separately in two subregions. We focus on emissions from 9 selected cities per subregion and present monthly averages and 4-year averaged emission maps at $0.04°$ grid size for a total number of 54 cities. The $NO_x$ emissions maps reveal unprecedented levels of detail for a large and diverse collection of cities. $NO_x$ emission hot spots are consistently found at large city cores, while some cities feature significantly higher emissions than others, most notably at the USA-Mexico border (San Diego vs. Tijuana and El Paso vs. Juarez). The spatial windows of some cities envelops much more prominent emission hot spots than the city cores, which generally correspond to large seaports and industrial areas. The average emissions in 2019–2022 are generally larger in East Asian cities, as 13 out of 18 cities in this region are higher than 10 nmol m$^{-2}$ s$^{-1}$. In contrast, only one city in the North American region (Tijuana) and two cities in the European region (Algiers and Istanbul) are higher than this value.

With respect to the temporal variation of $NO_x$ emissions, we choose the year of 2019 as the pre-COVID-19 baseline year, so the relative emission changes in 2020–2022 to 2019 indicate the post-COVID-19 perturbations to each city. We caution that the relative differences between post-COVID-19 months in 2020–2023 and the corresponding months in 2019 may exist even without COVID-19. These non-COVID-19 factors include the Military World Games impact in Wuhan and pre-existing long-term decreasing trend in many cities in the northern Europe subregion, as indicated by higher emission in 2018 than 2019 (Fig. 9). The initial impact during the first outbreak of COVID-19 in early 2020 can be found in most cities, but their paths

diverge afterwards. We average the monthly emissions for each city to annual mean emissions in 2019–2022 and group the normalized annual mean emissions into 4 clusters. All but one city in the North American region are grouped in cluster 1, which is characterized by the smallest emission reduction in 2020 ($-7\%$) and steady increase afterwards, resulting in a $+5\%$ increase in 2022. Limited representations of Latin America (Tijuana, Juarez, and Chihuahua), Africa (Algiers), and Middle East (Istanbul) are all located in cluster 1. Future studies might be meaningful to test if the emission changing pattern of cluster 1 is common in these regions. The other clusters (2–4) feature much larger emission reductions than cluster 1 and differ by how these reductions are distributed in 2020–2022. The European cities are generally in cluster 2 with the largest impact in 2020 ($-31\%$), whereas the East Asian cities are generally in cluster 3 and 4 with the largest impacts in 2022 ($-25\%$ and $-37\%$).

In this study we fit scale heights at monthly resolution and fit chemical lifetimes for each climatological month to strike a balance between the quality of the fitting results and temporal resolution. However, we assume spatially homogeneous scale heights and chemical lifetimes within each subregion. Considering that the fitting is conducted over cleaner locations where free tropospheric $NO_2$ subcolumn is expected to take a larger fraction of the tropospheric column, the fitted scale heights and chemical lifetimes are likely overestimated for urban areas. Additionally, the $NO_x$ chemical lifetime is highly nonlinear with respect to $NO_x$ concentration (Valin et al., 2013; Laughner and Cohen, 2019). Therefore, although some aspects of the fitted results are consistent with the expected spatial and temporal variation of PBL height and $NO_x$ chemical lifetime, we caution that the inverses of scale heights and chemical lifetimes are fundamentally linear fitting parameters and suggest against over-interpreting the results. Future investigations might be helpful to achieve higher spatial granularity and/or considering the dependencies of scale height and chemical lifetime on the column amount. We choose a constant $NO_x$:$NO_2$ ratio, given the emphasis of this study on relative emission changes and timeliness of emission estimation. An improved understanding of global $NO_x$:$NO_2$ ratio over the atmospheric columns through which satellite sensors integrate will likely enhance the quality of estimated $NO_x$ emissions.

This work presents observation-based $NO_x$ emission estimations over large areas (covering three major continents), with fine spatial resolution ($0.04°$, resolving intracity emission variations), high temporal resolution (monthly), and timely results (until 31 January 2023). The main focus of this work is the relative emission changes for each city in the pre- and post-COVID-19 years. The absolute emission values of one city compared to another and absolute estimates of emissions month-by-month would be subject to larger uncertainties than the relative values, given the assumptions and simplifications discussed above. We expect future evaluations of spatiotemporal variations of derived emissions against known emission rates of point sources and bottom-up emission inventories. The current work flow requires only TROPOMI level 2 data and the ERA5 reanalysis, both publicly available with global coverage, and the open-source Python algorithm (Sun, 2023). It is our hope that this tool will benefit future studies over more regions in the world and using additional remote sensing instruments.

*Code availability.* Code relevant to this paper can be found in Sun (2023).

*Data availability.* TROPOMI NO$_2$ PAL data are available at https://data-portal.s5p-pal.com/products/no2.html. TROPOMI NO$_2$ offline data are available at https://doi.org/10.5270/S5P-s4ljg54. The ERA5 data are available at https://doi.org/10.24381/cds.adbb2d47.

## Appendix A: Key assumptions in flux divergence vs. directional derivative approach

The flux divergence approach (e.g., Beirle et al., 2019, 2021; de Foy and Schauer, 2022; Dix et al., 2022) is based on the following equation, expressed in terms defined in this work.

$$\langle E \rangle = \langle \nabla \cdot (\Omega \boldsymbol{u}) \rangle + \frac{\langle \Omega \rangle}{\tau}$$
$$= \langle \boldsymbol{u} \cdot (\nabla \Omega) \rangle + \langle \Omega (\nabla \cdot \boldsymbol{u}) \rangle + \frac{\langle \Omega \rangle}{\tau}. \tag{A1}$$

Here the second step makes it clearer to compare with the counterpart of the directional derivative approach (i.e., Eq. 1). The key implicit assumptions of the flux divergence approach are discussed below.

1. The emission includes the divergence of horizontal flux and chemical loss. Without the chemical loss, the emission equals the horizontal flux divergence, as shown by studies applying flux divergence to methane (Liu et al., 2021; Veefkind et al., 2023). The problem is that the divergence of horizontal flux is also driven by the divergence of wind ($\nabla \cdot \boldsymbol{u}$), which can have positive or negative values climatologically for different locations. This leads to spurious emission values seen in the flux divergence literature that often need empirical correction (Liu et al., 2021; Dix et al., 2022; Veefkind et al., 2023).

2. The topography does not contribute to the flux divergence. In reality, the wind vector usually partially aligns with the gradient of surface altitude even over a long-term average, resulting in terrain-dependent artifacts.

The directional derivative approach (Sun, 2022, this work) addresses these assumptions by explicitly considering the wind divergence and topography effects. The assumptions that lead to the directional derivative approach are detailed in Sun (2022) and discussed below.

1. There exists an altitude $z_1$ where emissions, as observed by satellites, are confined within. We equate $z_1$ as the PBL height for ease of conceptualization, but it does not have to be explicitly defined to derive Eq. 1.

2. The horizontal gradient of subcolumn amounts above $z_1$ is negligible compared to that below $z_1$ at the spatial scale of adjacent satellite observations.

3. The vertical flux of observed species at $z_1$ is only due to divergence/convergence of wind below $z_1$ and is thus not sensitive to emissions. This assumption is a consequence of assumption 1 and the assumption that air flow is incompressible.

4. The scale height of the observed species is a constant through the domain. This is necessary to relate the surface concentration to the column amount in the topography term.

5. The column-integrated chemical lifetime of the observed species is a constant through the domain. This is necessary to simplify the chemical loss term, and it is the same for the flux divergence approach.

Assumptions 1–3 are from reasoning. We encourage future testing of these assumptions, presumably through high-quality model simulations. Assumptions 4–5 are apparently significant simplifications. The following two paragraphs discuss their implications.

The scale height is expected to be lower over polluted regions than clean regions. We fit the scale height over rough terrains in each subregion, which are inherently cleaner than the urban areas. Therefore, the scale height applied to urban areas is likely overestimated, and the topography term is hence underestimated as it scales with the inverse of scale height. Fortuitously, the urban areas are generally situated over flat terrains. The median value of monthly term $|\langle \Omega \boldsymbol{u_0} \cdot (\nabla z_0) \rangle|$ for all 54 cities averaged in each city is $1.3 \times 10^{-7}$ mol m$^{-1}$ s$^{-1}$. That means neglecting the topography effect resulting from a 1000 m scale height would only give rise to an emission error of $1.3 \times 10^{-10}$ mol m$^{-2}$ s$^{-1}$, which is below the noise floor. However, there are two caveats. First, this does not mean that the topography term is unimportant. It might be small over the flat city, but it is large over rough terrains that are close to many cities. Second, some emission sources do appear over rough terrains.

The column-integrated chemical lifetime is a complicated and challenging parameter to obtain. A wide range of values and strategies exist in the literature. Two main factors determines its value, the chemical lifetime within the PBL and the partition of column amounts in the PBL vs. in the free troposphere. The PBL chemical lifetime is highly nonlinear. In the "NO$_x$-limited" regime, it decreases with increasing NO$_x$, whereas in the "NO$_x$-suppressed" regime, the relationship is reversed. The range of variation is within a factor of two (Valin et al., 2013; Laughner and Cohen, 2019). The PBL vs. free troposphere partition may have a larger impact given the high urban-rural column amount contrast and significant free tropospheric contribution in the clean regions (Silvern et al., 2019). Overall, we expect the column-integrated lifetime determined over relatively clean regions to be higher than the true value over urban areas. This is also consistent with the longer lifetimes shown by Fig. 1 than literature values of urban PBL NO$_x$ lifetime. Consequently, the chemical loss term is likely underestimated in polluted regions.

As such, both topography and chemical loss terms are expected to be underestimated for NO$_x$ over urban areas. This under-correction is preferred to over-correction. Directions of future improvements include using model simulations to inform the spatiotemporal variations of scale height and lifetime and fitting more complex functions (e.g., as polynomial functions of column amount) of the scale height and lifetime. The current constant scale height and lifetime are just the special case of zeroth order polynomial. This will require even higher signal-to-noise ratio, more observations, and/or finer spatial resolution than TROPOMI.

## Appendix B: Comparisons between this work and its precursors

*Author contributions.* CRL and KS edited the manuscript. KS curated the algorithm and datasets.

**Table B1.** Considerations of physical and chemical processes by this work and previous studies. The flux divergence and directional derivative approaches are distinct by whether wind divergence is included or excluded.

| Study | Wind divergence | Topography | Lifetime | $NO_x$:$NO_2$ |
|---|---|---|---|---|
| Beirle et al. (2019) | Included | None | 4 h | 1.32 |
| Beirle et al. (2021) | Included | None | None | Photo-stationary state |
| Dix et al. (2022) | Included | Empirical background correction[a] | Calculated based on OH | 1.32 |
| de Foy and Schauer (2022) | Included | None | 9 h | 1.32 |
| Goldberg et al. (2022) | Included | None | Fitted using EMG[b] | 1.32 |
| Chen et al. (2023) | Included | None | Calculated using surface measurements | 1.32 |
| Sun (2022) | Excluded | Fitted monthly over the CONUS | Fitted over the CONUS after aggregating 2018–2022 | 1.32 |
| This work | Excluded | Fitted monthly over subregions with similar climate | Fitted over subregions with similar climate for each climatological month | 1.32 |

[a] This may compensate both topography and chemical loss effects. [b] EMG = exponentially modified Gaussian function.

*Competing interests.* The authors declare that they have no conflict of interest.

*Acknowledgements.* This research has been supported by the NASA Earth Science Division Rapid Response and Novel Research in Earth Science program (RRNES, Award 80NSSC20K1295) and Atmospheric Composition: Modeling and Analysis (ACMAP, Award 80NSSC19K0988). We thank Guanyu Huang, Hyeong-Ahn Kwon, Caroline Nowlan, and Heesung Chong for helping identifying emission hot spots.

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
