# Peer review of "Nitrogen oxides emissions from selected cities in North America, Europe, and East Asia observed by TROPOMI before and after the COVID-19 pandemic"

_EGUsphere, 2023_

## Author Comment (AC1)

**Responses to referee 1:**

We would like to thank the referee for the useful comments and constructive suggestions. In the following, we address the referee's comments and describe corresponding changes we have made to the manuscript. The referee's comments are listed in *italics*, followed by our response in blue. New/modified text in the manuscript is in **bold**.

*The paper by Lonsdale and Sun presents an interesting analysis of the effects of COVID-19 on NOx emissions in 54 cities over North America, Europe and East Asia, as well as the region- and city-specific features of such effects. The calculation is based on their previously developed "directional derivative approach" which considers the mass conservation under the influence of horizontal advection, emissions, chemical loss and terrain effect. The paper is well written and easy to read. I applaud their efforts to use a new emission inversion algorithm to study the COVID effects on air quality. There are a few major issues on the methodology and results in this paper that should be addressed.*

We thank the referee for recognizing the novelty of the directional derivative approach and strive to address the referee's concerns below.

*Based on Eq. 1 (and Sun et al., 2022, GRL), the "wind divergence" term $(\Omega \langle \nabla \cdot \vec{u} \rangle)$ is assumed to be equal to vertical flux at z1. This assumption might be overly simple. Consider a flat terrain, in which the terrain term becomes zero, Eq. 1 apparently misses the wind divergence.*

We thank the referee for bringing up the wind divergence term, which is the most critical difference between the directional derivative approach from this work and the recently established "flux divergence" approach (Beirle et al., 2019, 2021; de Foy and Schauer, 2022; Dix et al., 2022; Liu et al., 2021; Veefkind et al., 2023). We argue that the wind divergence term is not "missed", instead, it should be excluded, otherwise the divergence of horizontal wind vector will cause spurious positive/negative emissions which, depending on the value of $\Omega$, interferes with or completely overwhelms the true emission signals. Adopting the referee's suggestion of assuming flat terrain, and assuming no chemistry to focus on the transport terms, the flux divergence approach includes the wind divergence when calculating the emission:

$$\langle E \rangle = \langle \nabla \cdot (\Omega \vec{u}) \rangle$$
$$= \langle \vec{u} \cdot (\nabla \Omega) \rangle + \langle \Omega(\nabla \cdot \vec{u}) \rangle$$

The consequence of including the wind divergence term (highlighted by red in the equation above) is clearly demonstrated by Fig. 1a in Sun (2022), where large negative emissions are found over coastal land, positive emissions are found over coastal ocean, and the Gulf Stream appears as a strong sink due to its wind convergence. All of those are highly unrealistic and apparently not real sources/sinks. Intuitively, the wind divergence should not correlate with emission. On the contrary, the directional derivative term $\langle \vec{u} \cdot (\nabla \Omega) \rangle$ makes intuitive sense.

The gradient of column amount $\nabla\Omega$ should align with the horizontal wind vector $\vec{u}$ where there is emission, as higher column amount should appear downwind.

We add a new Appendix section to the manuscript to explicitly discuss assumptions. See the response below.

*In Sun et al. (2022), there are a few key assumptions for which the validity is not (well) tested. For example, assuming no horizontal gradient of NO2 above PBL to derive Eq. 3, assuming surface concentration to be derived from the column using a single inverse scale height across a large domain, assuming the wind divergence to be equal to vertical flux at z1 (Eq. 10), assuming the chemical loss in the PBL to be the same as the loss for the whole column (Eq. 12). All these assumptions are necessary to finally derive the "directional derivative approach", but these assumptions are subject to large uncertainties and should be tested rigorously to ensure the results for 54 cities here are robust. These tests are especially necessary, because there are few tests in Sun et al. (2022).*

We appreciate the referee's efforts to review the methodology in Sun (2022). To begin with, we would like to make two factual clarifications. First, Eq. 3 in Sun (2022) neglects the horizontal gradient of $NO_2$ subcolumn above the PBL at the spatial scale of satellite pixels, which is only 3–6 km for TROPOMI. The horizontal gradient above the PBL surely exists, e.g, from dispersion of lightning $NO_x$ or long-range transport of wildfire plumes, but the spatiotemporal scale of mixing in the free troposphere should be much larger than emissions in the PBL. The mixing time scale through the troposphere is $\sim 1$ month whereas it is $\sim 1$ day in the PBL (Brasseur and Jacob, 2017). More importantly, these gradients above the PBL happen over large areas and at varying locations and would be largely averaged out at monthly/annual scales in this study.

Second, Eq. 12 in Sun (2022) does not assume the chemical loss in the PBL to be the same as the loss for the whole column. That equation is copied here:

$$\frac{\Omega_b}{\bar{\tau}} = \frac{\Omega}{\tau}.$$

For $NO_x$, $\Omega$ is the tropospheric column, and $\Omega_d$ is the subcolumn from surface to $z_1$ and can be understood as the PBL subcolumn. Consider a relatively clean condition where $\Omega_d = 0.5\Omega$, i.e., the PBL and free troposphere have an equal subcolumn amount, the chemical lifetime over the troposphere, $\tau$, will be twice the value in the PBL, $\bar{\tau}$, i.e., $\tau = 2\bar{\tau}$. This should also be a major reason that $\tau$ in this study appears longer than what people unusually see for $NO_x$ lifetime; it represents the lifetime of the tropospheric column, not just the PBL and certainly not at the surface.

To rigorously test individual assumptions and quantify their uncertainties, one would need a high resolution, spatiotemporally comprehensive, and validated CTM simulation. That is beyond the scope of this study and a significant project by itself. However, we would like to emphasize that Sun (2022) already compared $NO_x$ emissions in four cities over the

CONUS (New York, Los Angeles, Dallas, and Houston) with the JPL chemical reanalysis, which includes global 3D transport and chemistry and assimilates many observations (Section 3.3 and Fig. 4 of that paper). We would consider these as more direct tests to the final results—the aggregated city-scale emissions.

We do agree that the assumptions leading to the directional derivative equation and the potential consequences should be more clearly stated. The following is included in the newly added Appendix A to discuss these assumptions and the (often implicit) assumptions in its precursor, the flux divergence approach:

**"The flux divergence approach (e.g., Beirle et al., 2019, 2021; de Foy and Schauer, 2022; Dix et al., 2022) is based on the following equation, expressed in terms defined in this work.**

$$\langle E \rangle = \langle \nabla \cdot (\Omega \vec{u}) \rangle + \frac{\langle \Omega \rangle}{\tau}$$

$$= \langle \vec{u} \cdot (\nabla \Omega) \rangle + \langle \Omega(\nabla \cdot \vec{u}) \rangle + \frac{\langle \Omega \rangle}{\tau}.$$

[revised manuscript text omitted]

*A few major assumptions are taken in addition. For example, the fixed NO : NO2 ratio, the fixed lifetime (for monthly climatology and for a large domain), and the fixed scale height (for a large domain). The NOx chemistry is highly nonlinear and its chemical lifetime varies greatly in space and time. The vertical mixing, PBLH and convection, which are related to the "scale height", also vary substantially in space and time. How the assumptions will affect your city-specific results need clarification. There exist fast algorithms in the literature that have considered varying NO : NO2 ratio, as well as concentration-dependent lifetime to account for nonlinear chemistry, and these studies should be discussed and/or compared.*

Please see the previous response about scale height and lifetime. Regarding the $NO_x$:$NO_2$ ratio, we have devoted section 3.3 to discuss this issue. Basically, we could not find a reliable, public, and global $NO_x$:$NO_2$ dataset that can match the timeliness and resolution of TROPOMI. This $NO_x$:$NO_2$ ratio will have to be column-integrated, so any error in the PBL/free troposphere partition will propagate to the resultant $NO_x$:$NO_2$ and directly to the emission estimates, even when the $NO_x$:$NO_2$ ratios were individually correct in the PBL and in the free troposphere.

We are not aware of any "fast" algorithms that satisfy our needs for $NO_x$:$NO_2$, chemical lifetime, and scale height. Emission estimation from the directional derivative approach only needs the open-source Python code and TROPOMI and ERA5 data, which are consistent across the globe and throughout the data periods and are available near real time. One does not need a computing cluster to run the algorithm. Adding model simulation to the approach negates its advantage of usability and timeliness, and opens up to more uncertainties due to chemical mechanism, transport, and emission inventory selection. The issue of emission inventories is especially critical due to their time lags and inconsistency among regions.

Therefore, we made choices based on the principle of parsimony. Despite these compromises, the levels of consideration on atmospheric physics and chemistry surpass previous works on

multiple fronts. For example, although the scale height and lifetime are fixed over the subregion and at climatological/calendar months, the spatiotemporal granularity is significantly improved over previous studies. We add another Appendix section in the revised manuscript to compare this work and previous ones:

**"Table B1: Considerations of physical and chemical processes by this work and previous studies. The flux divergence and directional derivative approaches are distinct by whether wind divergence is included or excluded.**

| Study | Wind divergence | Topography | Lifetime | $NO_x$:$NO_2$ |
|---|---|---|---|---|
| Beirle et al. (2019) | Included | None | 4 h | 1.32 |
| Beirle et al. (2021) | Included | None | None | Photostationary state |
| Dix et al. (2022) | Included | Empirical background correction[a] | Calculated based on OH | 1.32 |
| de Foy and Schauer (2022) | Included | None | 9 h | 1.32 |
| Goldberg et al. (2022) | Included | None | Fitted using EMG[b] | 1.32 |
| Chen et al. (2023) | Included | None | Calculated using surface measurements | 1.32 |
| Sun (2022) | Excluded | Fitted monthly over the CONUS | Fitted over the CONUS after aggregating 2018–2022 | 1.32 |
| This work | Excluded | Fitted monthly over subregions with similar climate | Fitted over subregions with similar climate for each climatological month | 1.32 |

[a] This may compensate both topography and chemical loss effects. [b] EMG = exponentially modified Gaussian function.**"**

*The resulting scale height is often very high (up to 5 km). It becomes difficult to interpret the exact physical meaning – it is too high for PBLH – and how it can be assumed constant across a large domain without causing major uncertainties.*

We thank the reviewer for pointing this out. First, the scale height is not necessarily driven by the PBLH. The scale height for CO derived from this approach is 5–7 km, and the scale height for methane is 7–9 km. When the vertical profile is more evenly distributed across the column, the scale height would approach the atmospheric scale height. Consider a clean PBL with enhanced free tropospheric $NO_x$ due to lightning or long-range transport. The scale height may even be higher than the atmospheric scale height.

Moreover, the PBLH in the southwest US is high and up to 4 km in summer months according to reanalysis. See Fig. 6 for spatial distribution and Fig. A1 for time series averaged over the entire southwest US climate region in Ayazpour et al. (2023). The less latent and more sensible heat flux at the surface in drier areas and the larger surface roughness in high mountain areas should contribute to the higher PBLH.

Furthermore, we second the comment by referee 2: *"My impression is that the height scale and the lifetime are at least partly numerical tuning parameters that have a loose connection with a physical interpretation. This would explain the particularly large values: the beta values are the inverse of the parameters, so large values suggest smaller than expected impact of the corresponding terms in the equations."* The values are just driven by the data by minimizing "ghost" positive/negative emissions where/when there should not be any. Although the scale height and lifetime in the fitting correspond to physical terms, we would not over-interpret the results. We expand the discussion about this in the second last paragraph in the conclusions and discussion section:

**"In this study we fit scale heights at monthly resolution and fit chemical lifetimes for each climatological month to strike a balance between the quality of the fitting results and temporal resolution. However, we assume spatially homogeneous scale heights and chemical lifetimes within each subregion. Considering that the fitting is conducted over cleaner locations where free tropospheric $NO_2$ subcolumn is expected to take a larger fraction of the tropospheric column, the fitted scale heights and chemical lifetimes are likely overestimated for urban areas. Additionally, the $NO_x$ chemical lifetime is highly nonlinear with respect to $NO_x$ concentration (Valin et al., 2013; Laughner and Cohen, 2019). Therefore, although some aspects of the fitted results are consistent with the expected spatial and temporal variation of PBL height and $NO_x$ chemical lifetime, we caution that the inverses of scale heights and chemical lifetimes are fundamentally linear fitting parameters and suggest against over-interpreting the results. Future investigations might be helpful to achieve higher spatial granularity and/or considering the dependencies of scale height and chemical lifetime on the column amount."**

*The derived lifetime is also too high (tens of hours or even more than 100 hours), much longer than what one would expect for a city. What is the exact meaning of this derived lifetime? Why is it OK to use this high value to obtain a reasonable estimate of emission? Since the lifetime for each sub-region and climatological month is constant, what is the impacts on the*

*derived interannual changes in emissions?*

See the previous responses about adding Appendix A to the revised manuscript and about the fitted scale height.

*The derived emissions often show very strange seasonal patterns, e.g., emissions in winter could be a few times those in summer. The only anthropogenic source of NOx that exhibits strong seasonality of residential heating, but this source only contributes a small fraction of total emissions for a city. Better interpretation and explanations of the resulting seasonality should be given.*

We thank the reviewer for pointing this out. We expand the discussion in Section 4.1 by adding $NO_x$:$NO_2$ ratio as another possible contributor to the seasonality:

**"Strong seasonal variations with higher emissions in winter months are observed in some cites (e.g., all cities in the northern subregion, Dallas, Houston, San Diego, and Juarez), which are inconsistent with flatter seasonalities often given by bottom-up emission inventories (Sun et al., 2021). These observed seasonal variations might be caused by seasonally varying artifacts, such as retrieval biases, vertical sensitivity of the retrieval at the surface, and the uncertainties in the wind vectors. In addition, because we use a global constant $NO_x$:$NO_2$ ratio, its seasonality that is unaccounted for will propagate to the $NO_x$ emission seasonality. One would expect higher PBL $NO_x$:$NO_2$ ratio in winter than summer, but in the summer relatively more $NO_x$ is in the free troposphere, where $NO_x$:$NO_2$ ratio is higher than the PBL (Seinfeld and Pandis, 2016). As a result, the exact impact of $NO_x$:$NO_2$ ratio on each city is inconclusive. However, we note that no clear seasonality can be identified in Tijuana, whereas the adjacent San Diego shows a much more prominent seasonal pattern. This is inconsistent with the potential impacts by the aforementioned factors, because they should have impacted the estimated city emissions similarly at such a close distance. Moreover, similar seasonalities are not so common in the regions of Europe and East Asia to be shown in the following sections. Further validation of the emission values and seasonality will be the subject of future studies."**

We also add the following sentences to last paragraph in the conclusion to be cautious about the absolute emission data:

**"The main focus of this work is the relative emission changes for each city in the pre- and post-COVID-19 years. The absolute emission values of one city compared to another and absolute estimates of emissions month-by-month would be subject to larger uncertainties than the relative values, given the assumptions and simplifications discussed above. We expect future evaluations of spatiotemporal variations of derived emissions against known emission rates of point sources and bottom-up emission inventories."**

*There are relatively detailed descriptions of the resulting emissions for each sub-region. How-*

*ever, it is not clear how the derived emissions are robust for each sub-region and city, given the abovementioned methodological weakness, and the fact that no independent data are used to compare with the emission results. There exist many social indices and other quantitative (proxy) data that can be used to represent the mobility change and lockdown stringency during COVID. Many studies of COVID have used these data, as should be used here.*

We did not compare with mobility change/lockdown stringency indices, as we did for Po Valley in Italy in Sun et al. (2021), for two reasons. First, cities included in this study cover a wide range of major $NO_x$ sources, such that we do not think mobility is representative collectively. Additionally, industrial regions and sea ports in many urban areas appear to be much larger than conventional "city" emissions. Second, the emission-derivation approach in this work uses globally consistent observation datasets and builds upon first principle-based equations. The inter-country/inter-continent consistency, data quality, and traceability of proxies and indices are not deemed robust enough to support the robustness of our derived emission. See the previous responses about weaknesses identified by the referee.

*The interpretation that the difference from 2019 is he effect of COVID should be cautious. For developing countries, their emissions often change from one year to another greatly, even without COVID, for example, due to economic growth and end-of-pipe control. Additional efforts should be made to assess (quantitatively or qualitatively) the non-COVID factors. There have been many COVID related studies that address this issue, for example, by taking advantage of the pre-COVID trend or variability.*

We thank the reviewer for this comment. Practically, the TROPOMI $NO_2$ record started in May 2018, so consistent data are limited (part of 2018 and 2019) to infer pre-COVID trend or variability. More importantly, COVID-19 has been constantly evolving in the studied regions. Every country and even every city have been impacted differently. Many COVID related studies that have been published focused on limited regions and/or short periods (usually months) and treated COVID as a one-off event. In reality, the COVID-19 impacts linger to this day and continue their diffusion to many aspects of human societies. We do not think it makes much sense to extrapolate the trends from 2019 and before to 2023 and to cut a boundary between "COVID" and "non-COVD" factors for each city. As such, we replace all appearances of "**COVID-19-induced variations/perturbations**" to "**post-COVID-19 variations/perturbations**" We also add the following sentence to the second paragraph of Section 4.2:

"**Some of the post-COVID-19 reductions relative to 2019 may extend from a pre-existing decreasing trend, as indicated by consistently higher 2018 emissions than 2019 in some cities in the northern subregion (Fig. 9).**"

And add the following to the second paragraph of the Section of conclusions and discussion:

[revised manuscript text omitted]

---

## Author Comment (AC2)

**Responses to referee 2:**

We would like to thank the referee for the useful comments and constructive suggestions. In the following, we address the referee's comments and describe corresponding changes we have made to the manuscript. The referee's comments are listed in *italics*, followed by our response in blue. New/modified text in the manuscript is in **bold**.

*This manuscript applies the flux divergence method to numerous cities on 3 continents in order to identify the changes in emissions over the 3 years impacted by COVID-19 lockdowns. The method includes refinements on the handling of terrain and lifetime that were reported in a prior publication. A clustering algorithm was applied to show that cities in North America, Europe and Asia had very different annual variability in NOx emissions over the last 4 years.*

*I believe that the method is sound and the results are valuable. The paper is clear and well written. I am happy to recommend it for publication.*

We appreciate the positive feedback from the referee. Strictly speaking, the method in this work does not calculate flux divergence. Therefore, it is more accurate to refer to the method as the "directional derivative" method. The newly added Appendix B compares this work and previous works applying the flux divergence method in details.

*My impression is that the height scale and the lifetime are at least partly numerical tuning parameters that have a loose connection with a physical interpretation. This would explain the particularly large values: the beta values are the inverse of the parameters, so large values suggest smaller than expected impact of the corresponding terms in the equations. I wonder if the question of the parameters would merit some more discussion and caveats in the analysis.*

Thanks for this comment. We expand the discussion about this in the second last paragraph in the conclusions and discussion section:

**"In this study we fit scale heights at monthly resolution and fit chemical lifetimes for each climatological month to strike a balance between the quality of the fitting results and temporal resolution. However, we assume spatially homogeneous scale heights and chemical lifetimes within each subregion. Considering that the fitting is conducted over cleaner locations where free tropospheric $NO_2$ subcolumn is expected to take a larger fraction of the tropospheric column, the fitted scale heights and chemical lifetimes are likely overestimated for urban areas. Additionally, the $NO_x$ chemical lifetime is highly nonlinear with respect to $NO_x$ concentration (Valin et al., 2013; Laughner and Cohen, 2019). Therefore, although some aspects of the fitted results are consistent with the expected spatial and temporal variation of PBL height and $NO_x$ chemical lifetime, we caution that the inverses of scale heights and chemical lifetimes are fundamentally linear fitting parameters and suggest against over-interpreting the results. Future investigations might be helpful to achieve higher spatial granularity and/or considering the dependencies of scale height and chemical lifetime on the column amount."**

*My other impression is that for each site the time series is robust in a relative sense. However, I think there are probably larger uncertainties in the absolute emission values of one city compared to another and of absolute estimates of emissions in the winter compared with the summer. Because the purpose of the paper is to look at lockdown-induced variability, I don't think this is a major problem. However, I do think it should be discussed to prevent over-interpreting the data. A more detailed comparison of emission totals by city with published emission inventories is beyond the scope of this study, but would be interesting in the future.*

We add the following sentences to last paragraph in the conclusion to be cautious about the absolute emission data:

**"The main focus of this work is the relative emission changes for each city in the pre- and post-COVID-19 years. The absolute emission values of one city compared to another and absolute estimates of emissions month-by-month would be subject to larger uncertainties than the relative values, given the assumptions and simplifications discussed above. We expect future evaluations of spatiotemporal variations of derived emissions against known emission rates of point sources and bottom-up emission inventories."**

**References**

Laughner, J. L. and Cohen, R. C.: Direct observation of changing NOx lifetime in North American cities, Science, 366, 723 LP – 727, https://doi.org/10.1126/science.aax6832, 2019.

Valin, L. C., Russell, A. R., and Cohen, R. C.: Variations of OH radical in an urban plume inferred from $NO_2$ column measurements, Geophysical Research Letters, 40, 1856–1860, https://doi.org/10.1002/grl.50267, 2013.

---

## Author Response (AR2)

**Responses to referee 1:**

We would like to thank the referee for the useful comments and constructive suggestions. In the following, we address the referee's comments and describe corresponding changes we have made to the manuscript. The referee's comments are listed in *italics*, followed by our response in blue. New/modified text in the manuscript is in **bold**.

*The authors have addressed most of my previous comments, by adding a few appendixes to clarify the features, assumptions and caveats of the method. One critical thing I would like to point out is that, the wind divergence term, which the authors argued to be unphysical, is indeed physical and necessary in order to ensure mass conservation. The physical meaning is that when there is wind divergence, the same volume of air parcel expands, and thus the density of NO2 decreases. This term, together with the density gradient term, makes sure the mass conservation of an air parcel when there are no temporal change of density (as assumed in the study), loss and emissions. The negative emissions are NOT a result of including the wind divergence term, but from other simplifications of the flux divergence method, including but not limited to the use of temporally (a few hours) averaged wind data. Therefore, although there maybe value of excluding the wind divergence term, it is certainly not because it is unphysical. Because exclusion of the wind divergence term is a foundational assumption of this study, the justifications and caveats must be clarified in the main text.*

We again appreciate the referee's efforts to scrutinize the directional derivative methodology established in Sun (2022), which contains detailed derivations, justifications, and evaluations in its main text and supporting information and has already undergone a rigorous peer review process of the journal *Geophysical Research Letters*. Although the goal of this current ACP manuscript is to apply the established directional derivative approach to unveil post-COVID emissions, instead of reinventing the methodology, we recognize that the readers who are interested in the details do not necessarily have to go back reading Sun (2022). As such, we add more clarification as the second paragraph of Section 3.2 of the manuscript, which is included at the end of this response. Before that, we would like to address the referee's comments specifically.

The referee asserted that "*the wind divergence term, which the authors argued to be unphysical*". It is unclear how the referee got this impression. We copy the relevant part in our previous response:

"*The consequence of including the wind divergence term (highlighted by red in the equation above) is clearly demonstrated by Fig. 1a in Sun (2022), where large negative emissions are found over coastal land, positive emissions are found over coastal ocean, and the Gulf Stream appears as a strong sink due to its wind convergence. All of those are highly unrealistic and apparently not real sources/sinks.*"

To paraphrase that, it is the inclusion of the wind divergence term that leads to unrealistic emissions (how can the Gulf Stream be a huge sink of trace gases larger than all known flux values?). We did not say that the wind divergence itself is unphysical. In fact, we did not

even use the word "unphysical".

We generally agree with the referee's following physical interpretation of the wind divergence term. To clarify, although the wind divergence term does not appear in the directional derivative equation (Eq. 1) in the manuscript, it does not simply vanish or get ignored. Instead, it is cancelled out by the vertical flux at $z_1$. Both Sun (2022) and the added Appendix A in the manuscript emphasized that the air flow is incompressible (it is compressible when flow becomes as fast as 0.3 Mach number)(Smits, 2000). As a result, if we take a column-shaped control volume from surface to $z_1$, the density of air in the control volume does not change, and the volume of the control volume, by construction, does not change. The horizontal wind divergence draws air down at $z_1$, and the horizontal wind convergence pushes air up at $z_1$. The following was stated in Appendix A in the previous revision, and we move these sentences to the main text:

**"Conceptually, the upward flux of the observed species at $z_1$ would not be due to emissions, as $z_1$ is chosen not to "feel" the emission impact; the only cause of this flux is the convergence of air in the column below that squeezes air upwards or the divergence of air below that draws air downwards."**

The equations that incorporate this and lead to the directional derivative equation are Eqs. 9–13 in Sun (2022). We summarize these in the newly added text to this manuscript at the end of this response.

The referee further alleged that *"The negative emissions are NOT a result of including the wind divergence term, but from other simplifications of the flux divergence method, including but not limited to the use of temporally (a few hours) averaged wind data"*, which we respectfully disagree. See the added Eq. A1:

$$\langle E \rangle = \left\langle \nabla \cdot (\Omega \vec{u}) \right\rangle + \frac{\langle \Omega \rangle}{\tau}$$
$$= \left\langle \vec{u} \cdot (\nabla \Omega) \right\rangle + \left\langle \Omega (\nabla \cdot \vec{u}) \right\rangle + \frac{\langle \Omega \rangle}{\tau}.$$

Ignoring the chemical term $\langle \Omega \rangle / \tau$, the emission calculated that way is just the sum of the wind divergence and directional derivative term. If the wind divergence is a large negative number, the calculated emission will be a similar large negative number. Additionally, we are unclear how the referee had the impression of the *"use of temporally (a few hours) averaged wind data"*, which we did not. Both Sun (2022) (Section 2) and this manuscript (Section 2.1) clearly state that the horizontal winds are spatiotemporally interpolated/sampled at TROPOMI level 2 observations from ERA5, where ERA5 winds are instantaneous at hourly resolution. We have thoroughly discussed the simplifications of both the flux divergence and the directional derivative approach in Appendix A, and none of these simplifications could explain the large, oscillatory positive/negative "emissions" calculated from the flux divergence, except the wind divergence term.

Finally, we summarize the clarifications and justifications above as the second paragraph of Section 3.2 of the revised manuscript:

"The most important difference between the flux divergence and directional derivative approach is, at flat surface and without chemical loss, whether the emissions equal the divergence of horizontal flux ($\langle \nabla \cdot (\Omega \vec{u}) \rangle$) or a directional derivative of the column amount ($\langle \vec{u} \cdot (\nabla \Omega) \rangle$). The mathematical and physical justifications of using the directional derivative instead of the flux divergence to estimate emission are provided in detail by Sun (2022), and we further list the key assumptions made by the flux divergence and directional derivative approaches in Appendix A. In brief, we assume an altitude $z_1$ that divides the lower troposphere where emissions are mixed within and the upper troposphere where emissions are not "felt" at the satellite pixel scale, and horizontal variability is much smaller than the lower part. Together with the incompressible flow assumption (Smits, 2000), these enable us to cancel out the wind divergence term from surface to $z_1$ ($\Omega_b(\nabla \cdot \vec{u})$, where $\Omega_b$ is the subcolumn from surface to $z_1$) with the vertical flux at $z_1$. Conceptually, the upward flux of the observed species at $z_1$ would not be due to emissions, as $z_1$ is chosen not to "feel" the emission impact; the only cause of this flux is the convergence of air in the column below that squeezes air upwards or the divergence of air below that draws air downwards. Ultimately, this leads to the only appearance of the directional derivative term in Eq. 1, instead of the flux divergence term that can be decomposed to the sum of the directional derivative term and a wind divergence term (see Eq. A1). Moreover, this study includes in general more advanced considerations of atmospheric physical and chemical processes in comparison with previous studies, which we summarize in Appendix B."

**References**

Smits, A. J.: A physical introduction to fluid mechanics, John Wiley & Sons Incorporated, 2000.

Sun, K.: Derivation of Emissions from Satellite-Observed Column Amounts and Its Application to TROPOMI NO2 and CO Observations, Geophysical Research Letters, 49, e2022GL101 102, https://doi.org/https://doi.org/10.1029/2022GL101102, 2022.